# Evolution of Omicron lineage towards increased fitness in the upper respiratory tract in the absence of severe lung pathology

Arthur Wickenhagen [1,5], Meaghan Flagg [1,5], Julia R. Port[1,2], Claude Kwe Yinda[1], Kerry Goldin[1], Shane Gallogly[1], Jonathan E. Schulz[1], Tessa Lutterman[1], Brandi N. Williamson[1], Franziska Kaiser [1], Reshma K. Mukesh[1], Sarah van Tol [1], Brian Smith[3], Neeltje van Doremalen [1], Colin A. Russell [4], Emmie de Wit [1,6] ✉ & Vincent J. Munster [1,6] ✉

The emergence of the Omicron lineage represented a major genetic drift in SARS-CoV-2 evolution. This was associated with phenotypic changes including evasion of pre-existing immunity and decreased disease severity. Continuous evolution within the Omicron lineage raised concerns of potential increased transmissibility and/or disease severity. To address this, we evaluate the fitness and pathogenesis of contemporary Omicron variants XBB.1.5, XBB.1.16, EG.5.1, and JN.1 in the upper (URT) and lower respiratory tract (LRT). We compare in vivo infection in Syrian hamsters with infection in primary human nasal and lung epithelium cells and assess differences in transmissibility, antigenicity, and innate immune activation. Omicron variants replicate efficiently in the URT but display limited pathology in the lungs compared to previous variants and fail to replicate in human lung organoids. JN.1 is attenuated in both URT and LRT compared to other Omicron variants and fails to transmit in the male hamster model. Our data demonstrate that Omicron lineage evolution has favored increased fitness in the URT.

The SARS-CoV-2 Omicron lineage first emerged in late 2021, and the BA.1 variant rapidly replaced prior circulating lineages of SARS-CoV-2[1]. The Omicron lineage was genetically and antigenically more distinct than previous variants, resulting in substantial evasion of neutralizing antibody responses and differences in pathogenesis[2,3]. Since then, ongoing evolution within the Omicron lineage has resulted in new Omicron variants with persistent antibody evasion and continuous optimization of ACE2 (angiotensin-converting enzyme 2) binding[4]. The highly mutated variant BA.2.86, representing another major genetic change, emerged in 2023 and was quickly detected in several countries, replacing the circulating XBB and EG.5.1 variants[5]. One additional

mutation in the spike (S) protein of BA.2.86 (L455S) gave rise to the JN.1 variant, which displayed reduced ACE2 affinity, indicating that the immune evasion capabilities come at the expense of receptor binding[6,7]. Nonetheless, JN.1 immune escape gave enough advantage to compensate for reduced receptor affinity compared to previous XBB and EG.5.1 variants and became the dominant variant in the human population[6,8].

Since the emergence of the Omicron lineage, it has been shown that these variants are less pathogenic in animal models and the human population compared to earlier variants of concern (VOC), with possible confounding factors of prior immunity in the human

[1]Laboratory of Virology, Division of Intramural Research, National Institute of Allergy and Infectious Diseases, National Institutes of Health, Hamilton, MT, USA. [2]Laboratory of Transmission Immunology, Helmholtz Centre for Infection Research (HZI), Braunschweig, Germany. [3]Rocky Mountain Veterinary Branch, Division of Intramural Research, National Institutes of Health, Hamilton, MT, USA. [4]Laboratory of Applied Evolutionary Biology, Department of Medical Microbiology, Academic Medical Center, University of Amsterdam, Amsterdam, The Netherlands. [5]These authors contributed equally: Arthur Wickenhagen, Meaghan Flagg. [6]These authors jointly supervised this work: Emmie de Wit, Vincent J. Munster. ✉e-mail: emmie.dewit@nih.gov; vincent.munster@nih.gov

population[9–12]. This reduced pathogenicity was accompanied by attenuated replication of Omicron variants in models of primary human lung epithelium, including organoids and explant cultures[13,14]. While circulation in humans continues to be dominated by these Omicron variants, immune pressures from a population with previous immunity to SARS-CoV-2 continue to drive antigenic changes and the emergence of new Omicron variants.[15,16] Most of these changes can be found in the S protein of SARS-CoV-2 which mediates receptor binding and membrane fusion with the host cell. The S protein is also the primary antigen targeted by antibody responses[17] and mutations in S have been shown to affect pathogenicity, transmissibility, and species tropism[18–20]. These changes suggest that continued intra-Omicron evolution likely influences the pathogenicity, transmissibility, and species tropism of these novel Omicron variants.

Here, we examined the phenotypic characteristics of four contemporary Omicron variants (XBB.1.5, XBB.1.16, EG.5.1, and JN.1) with regard to pathogenicity, transmissibility, and susceptibility in the Syrian golden hamster and in the human nasal and alveolar epithelium. To identify changes in ACE2 receptor usage across the whole breadth of intra-Omicron evolution, we employed a viral pseudotype system to assess the intra-Omicron changes in utilizing human and hamster ACE2 and TMPRSS2 dependency. Further, we assessed virus replication and disease signs in the upper and lower respiratory tract of hamsters and correlated the findings to model systems of the human upper and lower respiratory tract epithelium. Lastly, we used our established model of contact and airborne SARS-CoV-2 transmission in hamsters[21] to investigate differences in transmissibility of contemporary Omicron variants in the hamster. Our results provide novel insights into the intra-Omicron evolution and its increased fitness in the upper respiratory tract.

## Results

### Reduced TMPRSS2 dependency of contemporary Omicron variants

Despite successful replication of the Omicron variants in the human population, these SARS-CoV-2 Omicron variants initially showed an attenuated phenotype in several animal models[9,10]. To better understand the intra-Omicron differences, we first focused on the entry efficiency of a panel of Omicron variants representing the whole breadth of Omicron evolution. We analyzed a panel of Omicron spikes from BA.1 up to JN.1 in mediating pseudovirus entry while focusing on the contribution of TMPRSS2 or hamster ACE2 towards entry (Fig. 1 and Supplementary Fig. S1). To this end, we used a vesicular stomatitis

virus (VSV) based pseudovirus system to infect target A549 cells stably expressing human or hamster ACE2 in the presence of TMPRSS2 or only human ACE2 in the absence of TMPRSS2. This allowed us to specifically compare the contribution of TMPRSS2 and the difference between human and hamster ACE2 during entry of the different Omicron variants. While WA-1 showed a 128-fold increase in entry efficiency on cells expressing both human ACE2 and TMPRSS2 compared to ACE2 alone, the Omicron variants showed only a marginal influence of TMPRSS2 on their entry efficiency (Fig. 1A). This shows that Omicron variants maintain a reduced TMPRSS2 usage in comparison to WA-1 and Delta. These differences were not attributed to differential production of pseudoviruses or target cells as target cells showed stable expression of ACE2 and TMPRSS2 and each variant pseudotype was compared with itself on these target cells (Supplementary Fig. S1). Replacing human ACE2 with hamster ACE2 in the presence of human TMPRSS2 resulted in slightly reduced entry efficiencies of the Omicron pseudoviruses (Fig. 1B) but still demonstrated the potential of contemporary Omicron variants to continue to use hamster ACE2 for viral entry. Assessment of hamster ACE2 was completed in the presence of human TMPRSS2 to specifically look at the possibility of using hamster ACE2 (Fig. 1A).

### JN.1 variant shows attenuated disease in vivo

To evaluate the fitness of contemporary Omicron variants in vivo, we inoculated hamsters with four of the most recent Omicron variants XBB.1.5, XBB.1.16, EG.5.1, and JN.1, and assessed virus replication kinetics and pathogenicity. Hamsters were inoculated via the intranasal (I.N.) route with $10^4$ $TCID_{50}$ and monitored until euthanasia at 5 days post-inoculation (DPI). We measured SARS-CoV-2 shedding daily in oral swabs and found that JN.1 had a shorter shedding duration compared to the other three Omicron variants (Fig. 2A, B). No animals showed weight loss for the duration of the experiment (Supplementary Fig. S2A).

Next, we assessed virus replication in the upper and lower respiratory tract by quantifying viral RNA and infectious virus in nasal turbinates and lung tissues. All variants replicated in the upper respiratory tract, as demonstrated by the detection of high levels of genomic RNA and infectious virus (Fig. 2C, D). In contrast, in the lungs, we detected genomic RNA in all animals inoculated with all variants, but the infectious virus was only consistently detected in EG.5.1-infected animals (Fig. 2C, D). Lung virus titers in EG.5.1-infected hamsters peaked at lower levels compared to previous variants D614G, Alpha, and Delta (Supplementary Table 1)[22–24]. Furthermore, while

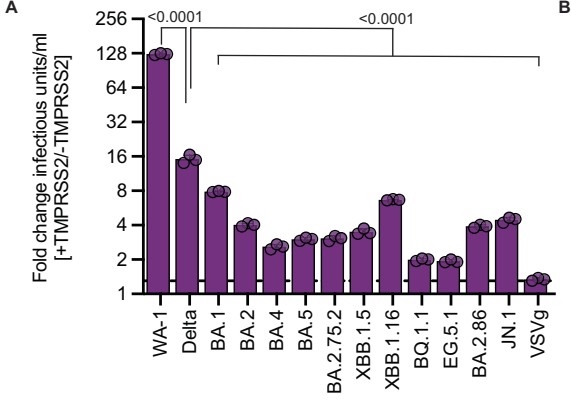
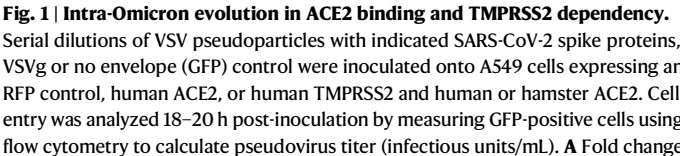
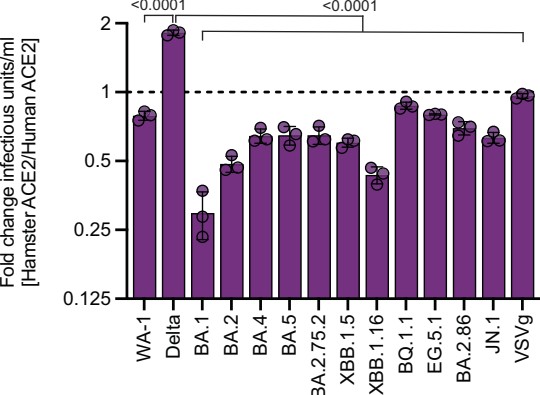

**Fig. 1 | Intra-Omicron evolution in ACE2 binding and TMPRSS2 dependency.** Serial dilutions of VSV pseudoparticles with indicated SARS-CoV-2 spike proteins, VSVg or no envelope (GFP) control were inoculated onto A549 cells expressing an RFP control, human ACE2, or human TMPRSS2 and human or hamster ACE2. Cell entry was analyzed 18–20 h post-inoculation by measuring GFP-positive cells using flow cytometry to calculate pseudovirus titer (infectious units/mL). **A** Fold change of pseudovirus titer with indicated spikes on cells expressing human ACE2 in the presence or absence of human TMPRSS2. **B** Fold change of pseudovirus titer with indicated spikes on cells expressing human TMPRSS2 in the presence of human or hamster ACE2. Data are plotted as mean +/− S.D. of $n = 3$ biological replicates. Statistical analysis was conducted using one-way ANOVA with Dunnett's post-test (**A, B**). P-values adjusted for multiple comparisons < 0.05 are shown.

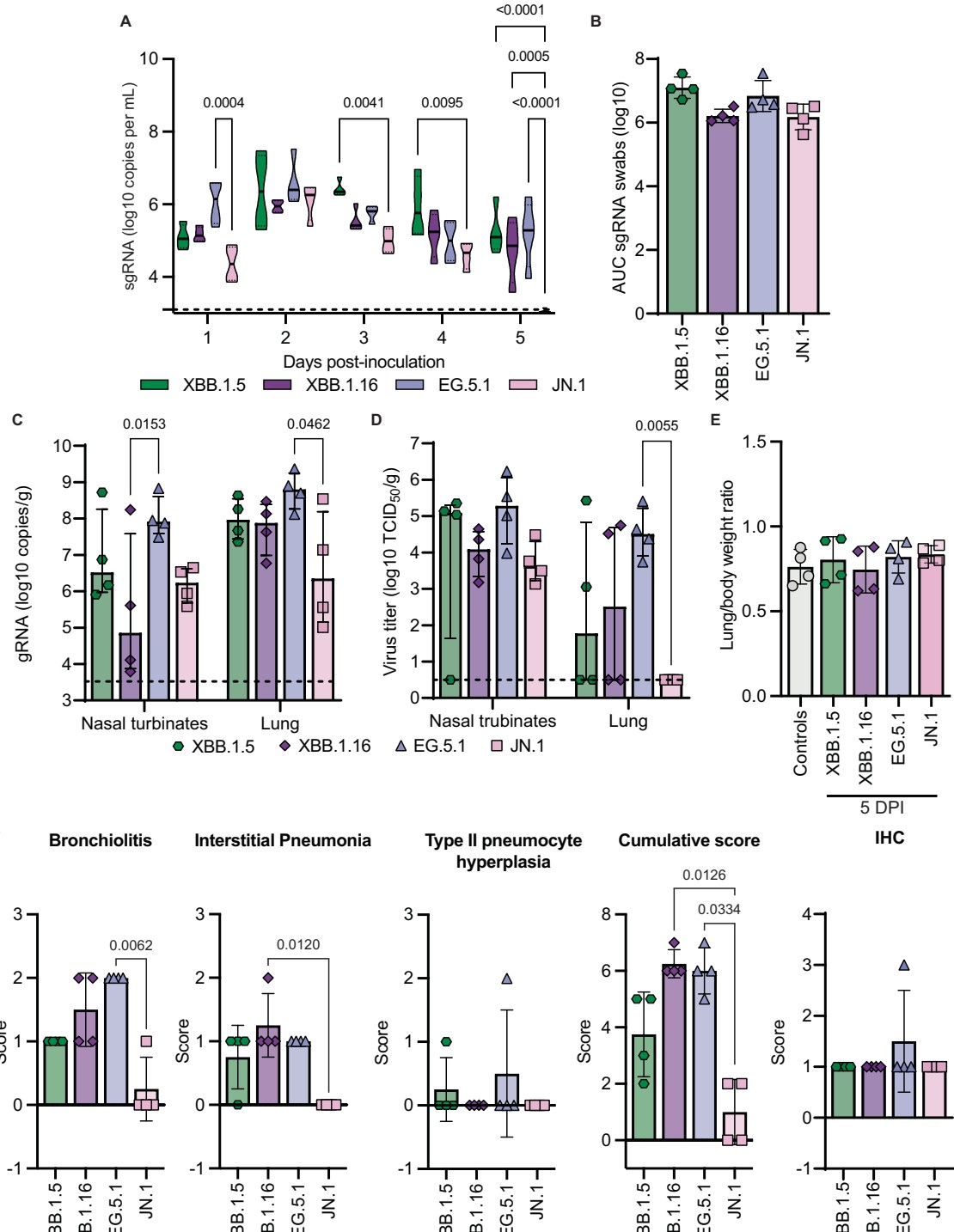

**Fig. 2 | Differences in virus replication kinetics in hamsters inoculated with Omicron variants.** Hamsters were inoculated intranasally with $10^4$ TCID$_{50}$ of indicated Omicron variants ($N = 4$ per variant). **A** Viral loads in oropharyngeal swabs were determined (sgRNA = SARS-CoV-2 subgenomic RNA). The dashed line indicates lower limit of quantification = 3.5 log$_{10}$ copies per mL. **B** Area under the curve (AUC) analysis of viral shedding as measured by sgRNA in oropharyngeal swabs from (**A**). Error bars denote mean +/− SD. **C, D** Virus replication in lungs and nasal turbinates of hamsters at 5 DPI. Viral loads (**C**) and virus titers (**D**) were determined. The dashed line indicates the lower limit of quantification = 3.5 log$_{10}$ copies per mL and the limit of detection = 0.5 log$_{10}$ TCID$_{50}$ per gram, respectively. Error bars denote the median with the interquartile range. **E** Lung/body weight ratio of hamsters at the time of euthanasia. Healthy, age-matched hamsters were used as controls. Error bars denote mean +/− SD. **F** Semi-quantitative scores were assigned to histopathological changes in the lungs of hamsters by a board-certified veterinary anatomic pathologist, as was the amount of virus antigen immunohistochemistry (IHC). Error bars denote mean +/− SD. Statistical analysis was conducted using two-way ANOVA followed by Tukey's post-test (**A**, **C**, **D**) or Kruskall-Wallis test followed by Dunn's multiple comparisons correction (**B**, **E**, **F**). *P*-values adjusted for multiple comparisons < 0.05 are shown.

infectious virus was occasionally detected in the lungs of XBB.1.5- and XBB.1.16-infected animals, JN.1 failed to replicate in the lungs after I.N. inoculation (Fig. 2D).

To assess damage to the lungs of infected hamsters, we weighed lungs at the time of euthanasia to establish a lung/body weight ratio. This lung/body weight ratio represents pulmonary changes after infection due to fluid buildup and immune infiltrate in the lungs with a higher ratio indicating increased damage to the lungs. Compared to healthy, age-matched historical control animals, none of the hamsters challenged I.N. with these Omicron variants showed an increased lung/body weight ratio (Fig. 2E), indicating an absence of severe LRT disease in these animals. Histopathological analysis of the lungs of Omicron-infected hamsters further confirmed an absence of significant pathology in the JN.1-infected hamsters (Fig. 2F and Supplementary Fig. S2B). While animals inoculated with XBB.1.16 and EG.5.1 showed the largest histopathological changes, with mild to moderate multifocal bronchiolitis and interstitial pneumonia, the overall scores showed a strongly reduced pathogenicity compared to earlier variants of SARS-CoV-2 in the hamster model[24]. All Omicron variants had similar small regions where viral antigen was detected by immunohistochemistry (IHC), affecting < 5% of the lung (Fig. 2F and Supplementary Fig. S2).

## Contemporary Omicron variants exhibit enhanced fitness in human nasal epithelium

To evaluate the fitness of Omicron variants in the human upper respiratory tract, we employed an air-liquid interface (ALI) culture model consisting of primary human nasal epithelium. The culture of these cells at ALI results in the differentiation of a polarized pseudostratified mucociliary epithelium that closely resembles nasal epithelium in vivo. Nasal ALI cultures were inoculated with the different Omicron variants on the apical side to recapitulate the natural route of exposure. Apical and basolateral samples were collected up to 96 h post-inoculation to quantify virus replication. All variants replicated to similar viral loads in culture supernatant by 72 h post-inoculation (Fig. 3A). Viral loads were higher in apical wash samples, and no major differences were observed between apical and basolateral replication kinetics. Notably, contemporary Omicron variants XBB.1.5, XBB.1.16, EG.5.1, and JN.1 replicated significantly faster in nasal epithelium compared to Delta, with at least a 1.5-log increase in genome copies at 24 h post-inoculation, and even compared to ancestral Omicron lineage BA.1. JN.1 exhibited slightly slower replication kinetics compared to its predecessors XBB.1.16 and EG.5.1 (Fig. 3B). In accordance with faster replication kinetics of contemporary Omicron variants, increased levels of sub-genomic SARS-CoV-2 RNA (sgRNA) were observed in cell lysates of contemporary Omicron-infected samples at 48 h post-inoculation, compared to Delta, while minimal differences were observed at 96 h post-inoculation (Fig. 3C).

To evaluate cell death in infected nasal epithelium, we quantified lactate dehydrogenase (LDH) release into culture supernatant over the course of infection as a measure of cytotoxicity. Infection with most variants did not result in cytotoxicity compared to mock-infected controls (Supplementary Fig. S3A). We detected a trend towards increased LDH levels in EG.5.1-infected cultures at later time points, but this did not reach statistical significance (Supplementary Fig. S3A). This elevated cytotoxicity may be associated with the rapid replication kinetics observed for EG.5.1 (Fig. 3B, C). To determine if Omicron variants impacted epithelial barrier integrity even in the absence of cytotoxicity, we measured trans-epithelial electrical resistance (TEER). We did not observe a pattern of a substantial decrease in TEER relative to a baseline that differed from mock-infected controls over the course of infection, indicating that infection with Delta or Omicron variants did not compromise epithelial barrier integrity (Supplementary Fig. S3B). Taken together, the enhanced virus replication in the absence of increased epithelial cell damage suggests ongoing adaptation to the upper respiratory tract for Omicron variants.

## Increased pathology upon intratracheal inoculation of Syrian hamsters with Omicron variants

While the four tested Omicron variants still replicate efficiently in the upper respiratory tract of hamsters, the absence of lower respiratory tract pathology and reduced replication in the lower respiratory tract are marked differences compared to earlier variants of SARS-CoV-2 (Supplementary Table 1)[23,25]. Initially, it was hypothesized that reduced replication due to altered entry of the virus resulted in this absence of pathology in the lower respiratory tract. Previously, we have shown that the route of administration is a key factor in overcoming the attenuated phenotype of Omicron BA.1 in the lower respiratory tract and that direct deposition of the virus into the lungs via intratracheal inoculation results in efficient replication[26]. To investigate the potential of these more contemporary Omicron variants to overcome the attenuated disease phenotype and induce lung pathology, we inoculated hamsters via the intratracheal (I.T.) route with $10^4$ TCID$_{50}$. To gain a better understanding of the disease progression and observe the suitability of this model for pre-clinical intervention testing, we euthanized animals at 3, 5, and 7 DPI. We observed shedding in oral swabs on 1, 3, 5, and 7 DPI for all variants (Fig. 4A, B). While shedding in the I.T.-inoculated hamsters was comparable with that in I.N.-inoculated hamsters (Figs. 2A and 4A), the overall amount of virus shed over time was lower in the I.T. inoculated hamsters (Figs. 2B and 4B). As was observed after I.N. inoculation, less JN.1 virus was shed and for a shorter period after inoculation compared to the other three Omicron variants (Fig. 4A, B). While no weight loss was observed for JN.1-inoculated hamsters, the other Omicron variants induced minor weight loss in hamsters around 5 DPI while regaining their weight by 7 DPI (Supplementary Fig. S4A).

We further examined virus replication in the upper and lower respiratory tract of I.T. inoculated hamsters (Fig. 4C–F). Viral loads in the lungs were high at all time points with the highest levels detected at 3 DPI (Fig. 4C).[22–24] Consistently, similar levels of replicating virus in lung tissue at 3 DPI were detected in animals inoculated with XBB.1.5, XBB.1.16, and EG.5.1, with slightly lower levels observed for JN.1, and similar levels for all Omicron variants at 5 DPI (Fig. 4D). Notably, lung virus titers of Omicron I.T.-inoculated animals were still lower than those observed following I.N. inoculation with earlier variants D614G, Alpha, and Delta (Supplementary Table 1) A more nuanced picture was observed in the upper respiratory tract. SARS-CoV-2 genomic RNA was detected in the nasal turbinates at all timepoints for all Omicron variants, but JN.1 displayed lower levels at 3 DPI while EG.5.1 showed higher levels at 7 DPI compared to the other variants (Fig. 4E). This result was further confirmed by detection of infectious virus which showed that JN.1 migrated into the upper respiratory tract less efficiently, evidenced by only 1 out of 4 animals having detectable infectious virus at 3 DPI (Fig. 4F). In contrast, EG.5.1 showed the highest levels of virus in the upper respiratory tract and had detectable infectious virus at 7 DPI while most of the other variants did not (Fig. 4F).

We assessed the lung/body weight ratio at the time of necropsy as a measure of lung damage. All Omicron variants showed a lung/body weight ratio comparable with healthy, age-matched historical control animals, at 3 DPI (Fig. 4G). At 5 DPI, and less so at 7 DPI, XBB.1.5, XBB.1.16, and EG.5.1 showed an increased lung/body weight ratio consistent with SARS-CoV-2 induced lung pathology at the time of necropsy (Supplementary Fig. S4B, C). This is in contrast to I.N.-inoculated hamsters (Fig. 2E), demonstrating that I.T. inoculation with contemporary Omicron variants results in increased lung pathology, as observed for BA.1[26]. Again, JN.1 was the outlier of the Omicron variants with no increase in lung/body weight ratio at any time post-inoculation, indicating an absence of lung damage even after I.T. inoculation. Histopathological examination of the lungs confirmed the increased lung damage seen following I.T. inoculation, characterized by multifocal moderate bronchiolitis, interstitial pneumonia, and type II pneumocyte hyperplasia, with the peak of disease and initiation of

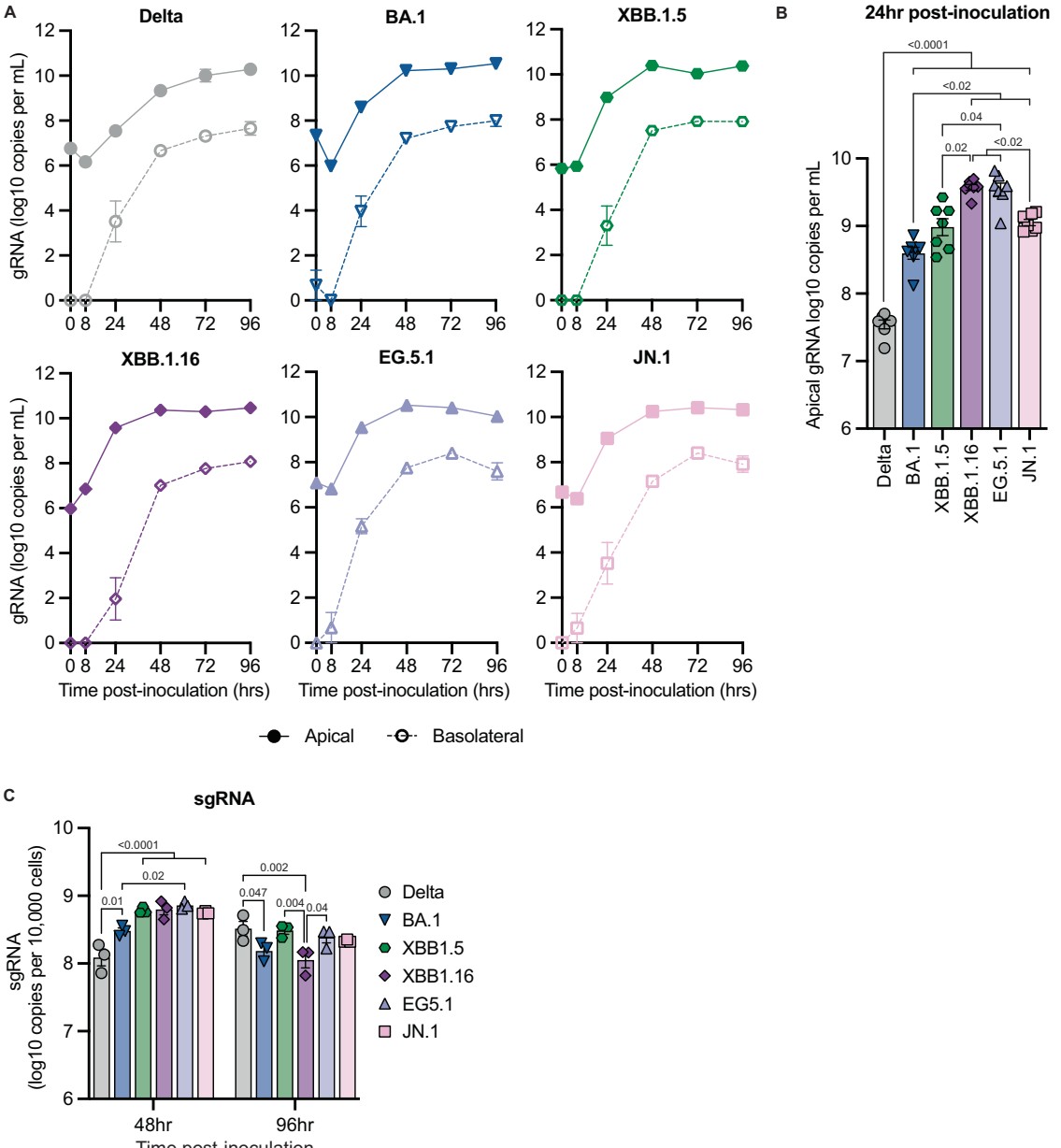

**Fig. 3 | Omicron variants replicate faster in human nasal epithelium compared to Delta.** Nasal ALI cultures were infected with SARS-CoV-2 variants at MOI 0.1 via the apical surface and serially sampled for 96 h. **A** Viral load in apical and basolateral culture supernatant was measured by qRT-PCR for SARS-CoV-2 gRNA. A dilution series of RNA standards with known copy numbers was run in parallel to calculate copy numbers. **B** Comparison of viral load in apical supernatant at 24 h post-inoculation, as measured in (**A**). **C** Viral load in cell lysate was quantified by qRT-PCR for SARS-CoV-2 sgRNA. A dilution series of RNA standards with known copy numbers was run in parallel to calculate copy numbers. Error bars denote mean +/– SEM of (**A**, **B**) $n = 7$ (0–48 h) $n = 4$ (72–96 h), or $n = 3$ (**C**) biological replicates. Statistical analysis was conducted using a restricted maximum likelihood mixed effects model (**B**) or two-way ANOVA (**C**), both followed by Tukey's post-test. $P$-values adjusted for multiple comparisons < 0.05 are shown.

lung repair observed at 5 DPI for all Omicron variants (Fig. 4H and Supplementary Fig. S4B, C). XBB.1.16 induced the most severe pulmonary pathology compared to other variants at all time points. IHC positivity for SARS-CoV-2 peaked at 3 DPI for all Omicron variants (Fig. 4I). JN.1 showed the lowest scores for bronchiolitis and interstitial pneumonia as well as overall pathology scores compared with the other Omicron variants (Supplementary Fig. S4B), indicating reduced pathogenicity of JN.1.

## Omicron variants are severely attenuated in human alveolar epithelium
To evaluate the replicative fitness and pathogenic potential of Omicron variants in the human lower respiratory tract, we infected

induced pluripotent stem cell- (iPSC) derived human alveolar organoids (ihLOs), which consist of type II pneumocytes. We generated ihLOs from human iPSCs according to established methodology[27]. We identified type II pneumocytes in ihLOs using transmission electron microscopy to visualize lamellar bodies, a hallmark feature of these cells in vivo (Supplementary Fig. S5A). In contrast to the nasal epithelium, replication of Omicron variants was minimal in ihLOs. Significant production of infectious virus only occurred in Delta- and XBB.1.16-infected ihLOs (Fig. 5A). Accordingly, increased sgRNA viral load in cell lysate was only detected in Delta- and XBB.1.16-infected ihLOs (Supplementary Fig. S5B). Increasing levels of viral gRNA over time were consistently detected in Delta- and XBB.1.16-infected ihLOs (Supplementary Fig. S5C). At 48 h post-inoculation, the virus titer in

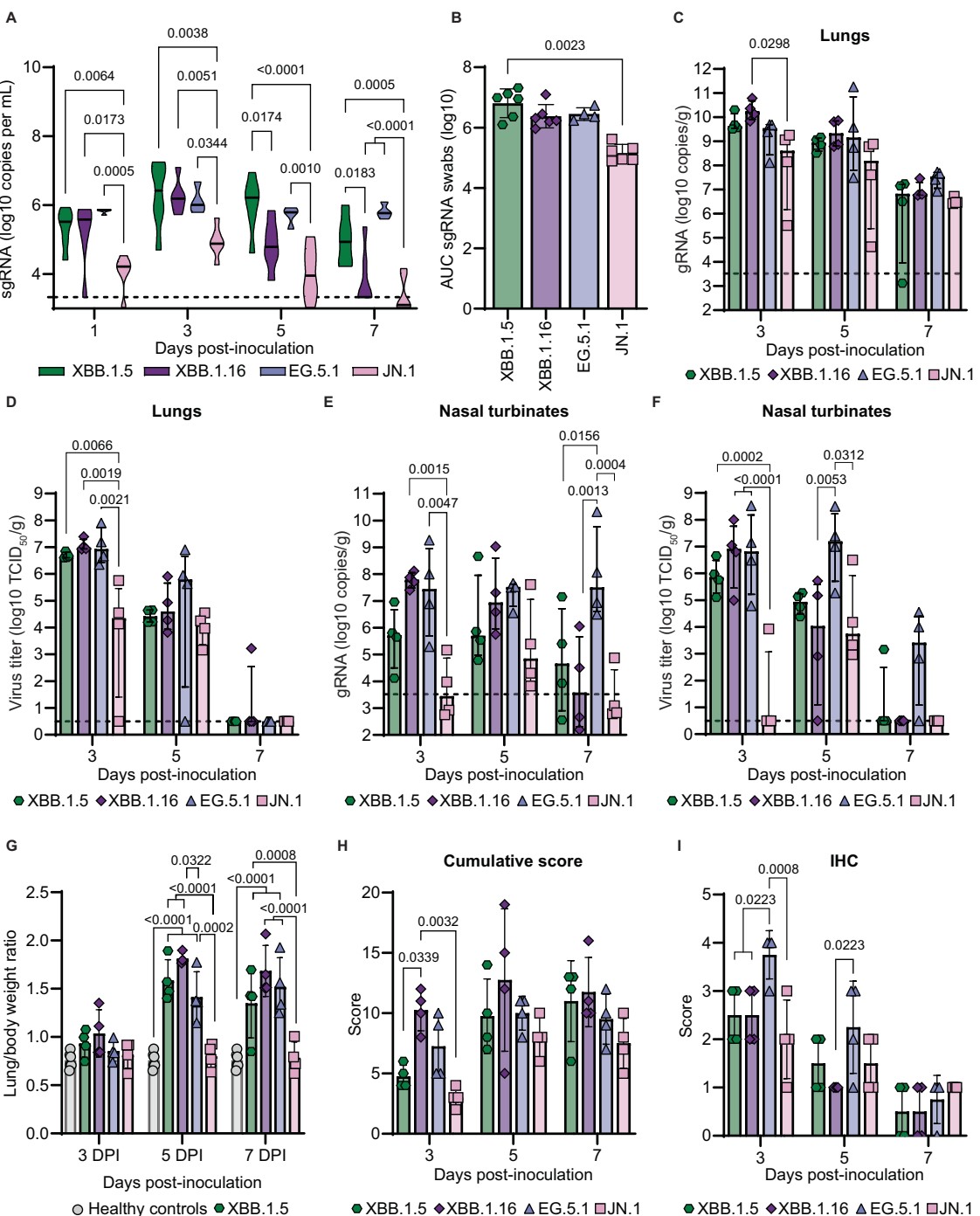

**Fig. 4 | Intratracheal inoculation of Syrian hamsters with Omicron variants results in increased pathology.** Hamsters were inoculated intratracheally with $10^4$ TCID$_{50}$ of indicated Omicron variants. **A** Viral loads in oropharyngeal swabs were determined (sgRNA = SARS-CoV-2 subgenomic RNA) at day 1, 3, 5, and 7 DPI. The dashed line indicates a lower limit of quantification = 3.5 log$_{10}$ copies per mL. **B** Area under the curve (AUC) analysis of viral shedding as measured by sgRNA in oro-pharyngeal swabs from (**A**). Error bars denote mean +/− SD of $N = 6$ per variant, except EG.5.1 $N = 4$. **C**−**F** Virus replication in lungs and nasal turbinates of hamsters at 3, 5, and 7 DPI. Error bars denote the median with interquartile range, $N = 4$ per variant and timepoint. Viral loads (**C**, **E**) and virus titers (**D**, **F**) were determined. The dashed line indicates the lower limit of quantification = 3.5 log$_{10}$ copies per mL and

the limit of detection = 0.5 log$_{10}$ TCID$_{50}$ per gram, respectively. **G** Lung/body weight ratio of hamsters at the time of necropsy. Healthy, age-matched hamsters were used as controls. Error bars denote mean +/− SD of $N = 4$ per variant and timepoint. **H**, **I** Semi-quantitative scores were assigned to histopathological changes in the lungs of hamsters by a board-certified veterinary anatomic pathologist, as was the amount of virus antigen immunohistochemistry (IHC). Error bars denote mean +/− SD of $N = 4$ per variant and timepoint. Statistical analysis was conducted using two-way ANOVA followed by Tukey's post-test (**A**, **C**−**I**) or Kruskall-Wallis test followed by Dunn's multiple comparisons correction (**B**). $P$-values adjusted for multiple comparisons < 0.05 are shown.

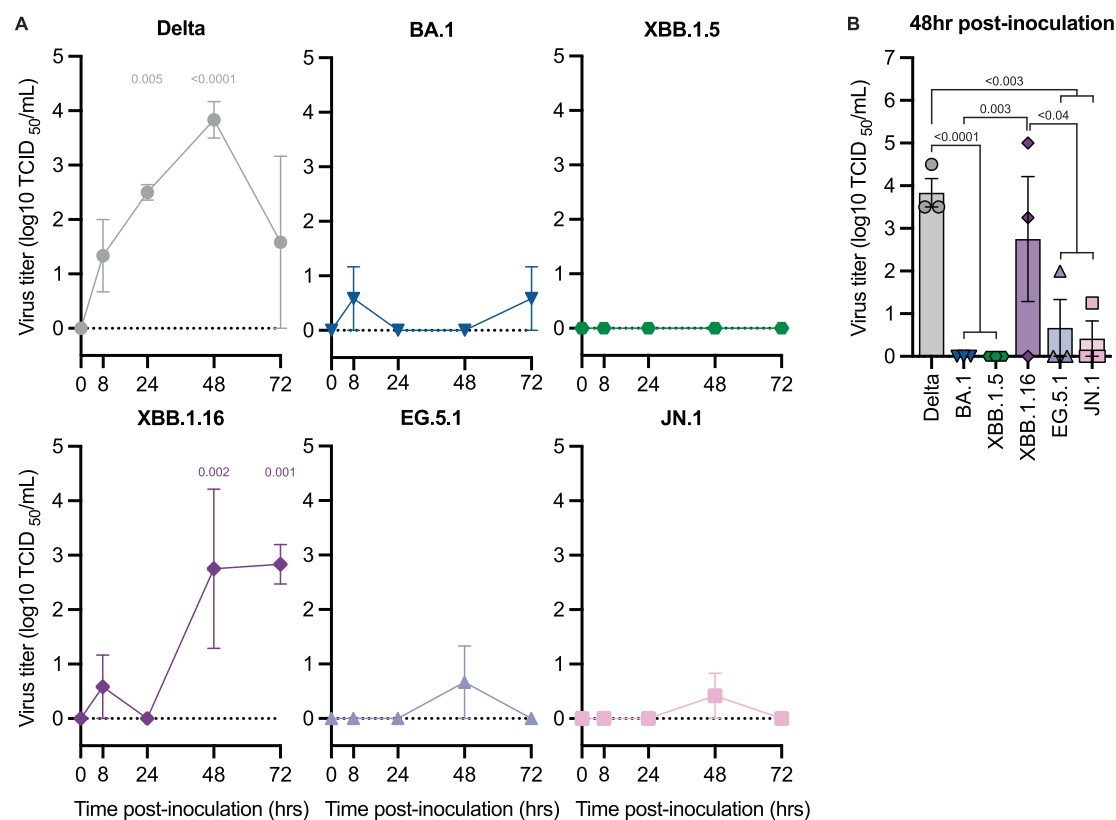

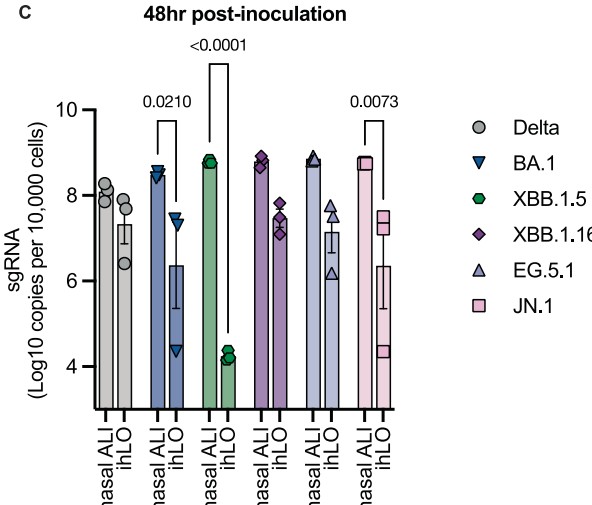

**Fig. 5 | Omicron variants are attenuated in iPSC-derived human alveolar organoids (ihLOs).** ihLOs were infected with SARS-CoV-2 variants at MOI 0.1. **A** Quantification of virus titer in culture supernatant. *P*-values denote difference vs 0 h post-inoculation. **B** Comparison of virus titer in culture supernatant at 48 h post-inoculation. **C** Comparison of sgRNA copies in cell lysate at 48 h post-inoculation in nasal ALI and ihLO cultures. Error bars denote mean +/− SEM of *n* = 3 biological replicates. Statistical analysis was conducted using two-way ANOVA followed by Dunnett's (**A**), Tukey's (**B**), or Sidak's (**C**) post-test. *P*-values adjusted for multiple comparisons < 0.05 are shown.

the supernatant of Delta- and XBB.1.16-infected ihLOs were significantly higher than other Omicron variants (Fig. 5B). These data are consistent with the fact that more severe pathology was observed in XBB.1.16 I.T.-inoculated hamsters and with previous reports that Omicron BA.1 is significantly attenuated in human lower respiratory tract epithelium[13,14]. We evaluated the pathogenicity of Omicron variants in ihLOs by measuring cytotoxicity via LDH release assay. Elevated cytotoxicity compared to mock-infected controls was not observed in ihLOs infected with any variant (Supplementary Fig. S5D), in agreement with previous reports[13]. The lack of virus replication in

ihLOs suggests that contemporary Omicron variants, except for XBB.1.16, have not significantly regained replicative fitness or pathogenicity in human alveolar type 2 pneumocytes.

We normalized detected sgRNA copies on a per-cell basis in order to compare replication kinetics of the different variants between nasal ALI cultures and ihLOs. While Delta replicated at similar levels in both model systems, viral loads of Omicron variants trended higher in nasal ALI cultures, with BA.1, XBB.1.5, and JN.1 reaching significantly higher viral loads in nasal ALI cultures compared to ihLOs at 48 h post-inoculation (Fig. 5C). This indicates that Omicron variants have

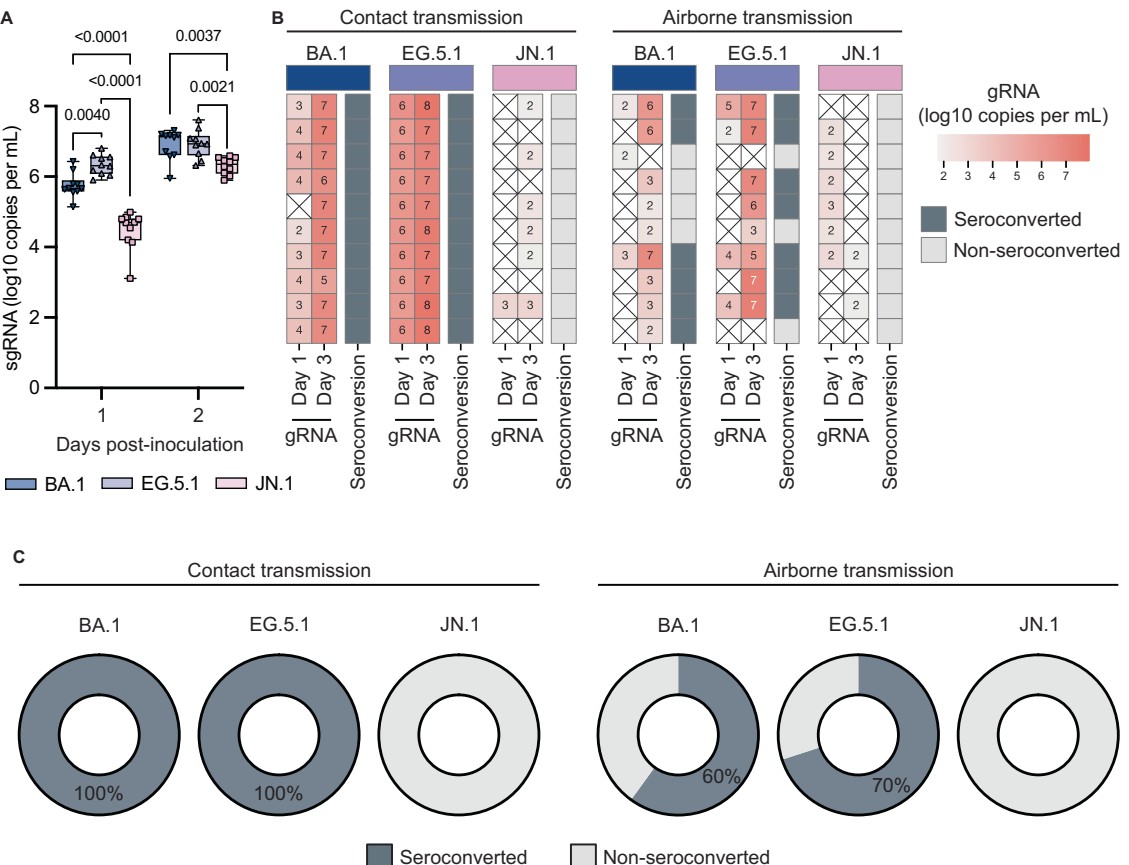

**Fig. 6 | JN.1 fails to transmit between hamsters via contact and airborne transmission.** Hamsters were inoculated intranasally with $10^4$ TCID$_{50}$ of indicated Omicron variants, $N = 10$ per variant. 36 h post-inoculation hamsters were co-housed for 24 h with one contact and one air-sentinel hamster before being single-housed for the remainder of the experiment. **A** Viral load was determined by subgenomic RNA in oropharyngeal swabs on days 1 and 2 post-inoculation of donor hamsters inoculated intranasally with $10^4$ TCID$_{50}$. Box defines the upper (75th percentile) and lower (25th percentile) quartiles with whiskers extending from minimum to maximum with all values shown. **B** Viral load was determined by genomic RNA in oropharyngeal swabs of sentinel hamsters at day 1 and 3 post-exposure with each box representing an individual sentinel hamster. Limit of quantification = 3.5 $\log_{10}$ copies per mL. Colors refer to the legend on the right. Seroconversion of sentinel hamsters was determined at day 14 post-exposure by ELISA. **C** Percentages indicate the fraction of seropositive sentinel hamsters as determined on day 14 post-exposure by ELISA. Statistical analysis was conducted using two-way ANOVA followed by Tukey's post-test (**A**). $P$-values adjusted for multiple comparisons $< 0.05$ are shown.

evolved an enhanced ability to replicate in nasal epithelium when compared with Delta.

## JN.1 fails to transmit between hamsters via contact and airborne transmission

The intra-Omicron evolution clearly showed changes in the replication kinetics and pathogenicity of the different Omicron variants. In hamsters, the biggest difference in replication kinetics of the tested Omicron variants was displayed between EG.5.1 and JN.1 variants. Therefore, we were interested to examine if these differences between the variants also resulted in changes in transmissibility between the viruses. Using 10 transmission pairs per Omicron variant, we compared contact and airborne transmission of Omicron variants BA.1, EG.5.1, and JN.1 in the hamster model. We selected BA.1 as a comparator, since this variant exhibits contact transmission but limited airborne transmission[26], and thus would allow us to detect a potential increase in airborne transmission for later Omicron variants. Donor hamsters were inoculated I.N. with $10^4$ TCID$_{50}$ of the respective Omicron variants and subsequently co-housed with one contact and one airborne sentinel from 36–52 h post-inoculation. After the contact time, all hamsters were single-housed, sentinels were swabbed on day 1 and 3 post-exposure and seroconversion was assessed at day 14 post-exposure. Consistent with our I.N. data (Fig. 2), donor animals showed detectable virus in oral swabs with JN.1 displaying reduced levels compared to

BA.1 and EG.5.1 (Fig. 6A). All contact sentinels for BA.1 and EG.5.1 showed detectable virus in oral swabs and seroconverted. EG.5.1 sentinels showed higher levels of viral genomic RNA and more hamsters tested positive on day 1 post-exposure compared to BA.1, with similar levels being detected on day 3 post-exposure (Fig. 6B). Airborne transmission of BA.1 resulted in detection of SARS-CoV-2 genomic RNA in 5 out of 10 animals on day 3 post-exposure and 6 out of 10 animals seroconverted. Airborne transmission of EG.5.1 was similar with 7 out of 10 animals testing positive for SARS-CoV-2 genomic RNA on day 3 post-exposure and 7 out of 10 animals seroconverted. Major differences were visible for both contact and airborne transmission for JN.1 (Fig. 6C). While some animals showed minor signals below the limit of quantification for genomic RNA, none of the JN.1 contact or air sentinels seroconverted by 14 days (Fig. 6B, C). This indicates that JN.1 has a vastly different transmission profile compared to Omicron BA.1 and EG.5.1 in the hamster model.

## Distinct immune evasion patterns of Omicron variants

The observed reduction of JN.1 fitness and transmission coincide with substantial genetic divergence and antigenic changes, likely due to immune pressure in a non-naive population. To confirm this, we collected serum samples 28 DPI from I.T. inoculated hamsters with resolved Omicron infection (Fig. 4) and assessed the antigenic distance between the Omicron variants in a virus neutralization assay using the

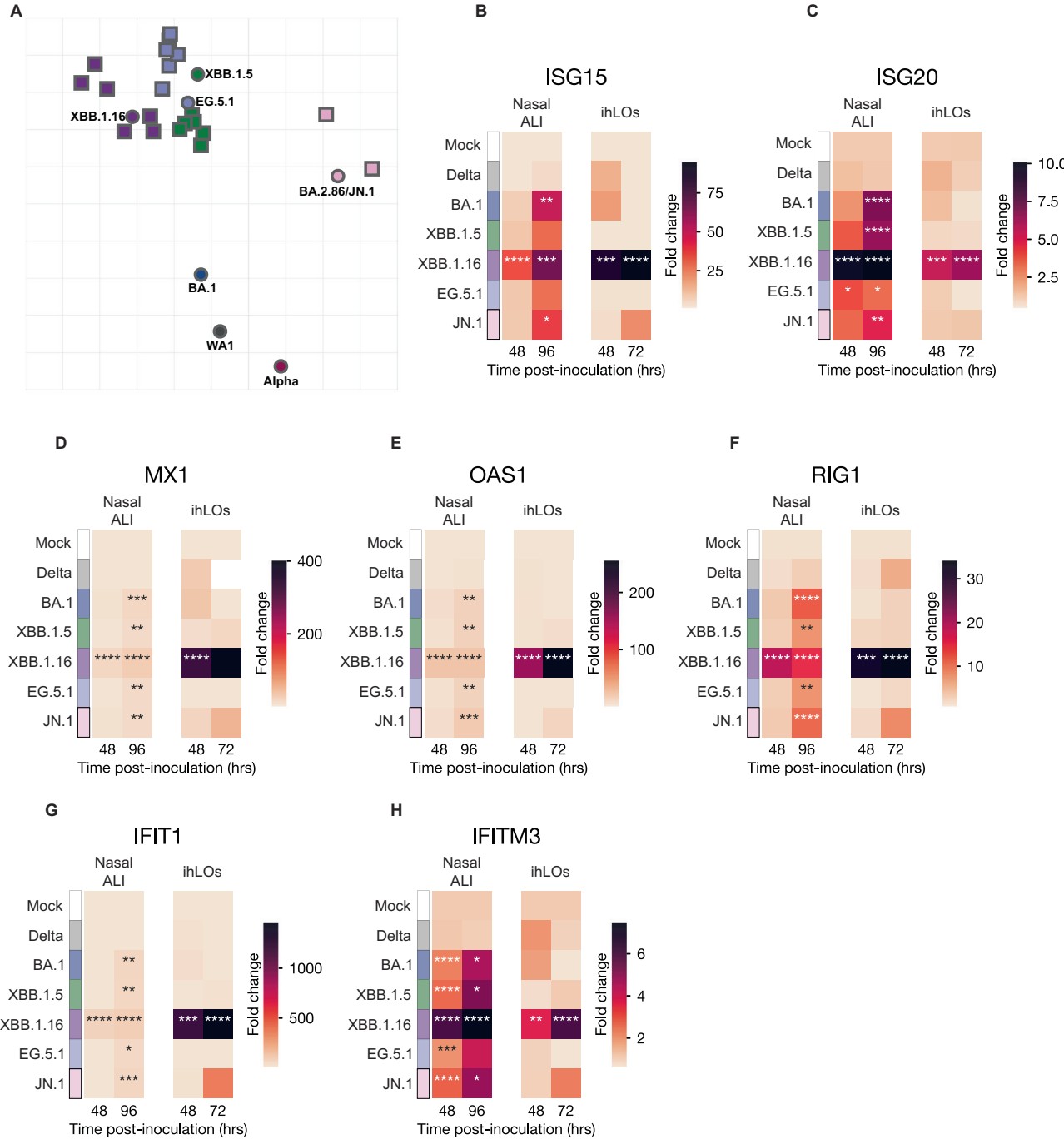

**Fig. 7 | Contemporary Omicron variants exhibit distinct patterns of antigenicity and innate immune activation. A** Antigenic cartography map demonstrating substantial antigenic differences between Omicron variants. Multidimensional scaling was used to show the antigenic distance between different antigens and sera obtained from I.T. inoculated hamsters, based on live VN titers. Hamster sera are shown in squares, and antigens are shown in circles. Most JN.1 sera could not be included because of insufficient non-zero VN titers. **B–H** Expression of IFN-stimulated genes in SARS-CoV-2 infected nasal ALI cultures and ihLOs was measured by qRT-PCR. Fold change in expression was calculated relative to timepoint-matched mock-infected controls. Samples were analyzed at 48 h post-inoculation and at the experimental endpoint (96 h post-inoculation for nasal ALI cultures, and 72 h post-inoculation for ihLOs). The mean fold change of $n \geq 3$ replicates is shown. Statistical analysis was conducted using one-way ANOVA with Dunnett's post-test. *P*-values adjusted for multiple comparisons < 0.05 for comparisons versus mock-infected controls are indicated. **** < 0.0001, *** < 0.001, ** < 0.01, * < 0.05.

same four Omicron variants and additional antigenically distinct variants Omicron BA.1, Alpha and WA-1 to capture the breadth of SARS-CoV-2 evolution (Supplementary Table 2). We then used antigenic cartography to assess the antigenic distance (Fig. 7A). The XBB.1.5, XBB.1.16, and EG.5.1 variants are similar to each other but distinct from other Omicron and SARS-CoV-2 variants. However, JN.1 is antigenically distinct from both XBB lineages and EG.5.1 as well as early BA.1

Omicron variants. This antigenic distance is further confirmed by the fact that most JN.1 sera could not be included because they cannot be reliably positioned within the map due to insufficient cross-neutralization against the other variants (Supplementary Table 2).

To determine if the Omicron variants evade innate immune responses as well as adaptive immunity, we analyzed the interferon (IFN) response in human nasal ALI and ihLO cultures infected with

Omicron variants. The induction of seven IFN-stimulated genes (ISGs) was measured by qRT-PCR. In both nasal ALI and ihLO cultures, infection with XBB.1.16 induced a significantly stronger ISG response compared to other variants (Fig. 7B–H). The lack of ISG induction in ihLOs infected with Omicron variants other than XBB.1.16 may be due to a lack of replicating virus (Fig. 5A). These differences partially normalized in nasal ALI cultures at the experimental endpoint (96 h post-inoculation) but were maintained in ihLO cultures throughout the experiment (72 h post-inoculation). Notably, we did not detect any induction of ISGs in response to infection with Delta despite detecting virus replication in both ihLOs and nasal ALI cultures, highlighting the enhanced immune evasion capabilities of this variant[28,29]. We also measured the induction of pro-inflammatory cytokines by qRT-PCR in nasal ALI cultures. Similar to the ISGs, significant induction of cytokine gene expression was only observed in Omicron-infected samples (Supplementary Fig. S6).

## Discussion

The emergence of the Omicron lineage represented a major genetic and antigenic change in SARS-CoV-2 evolution. This resulted in numerous phenotypic changes, including evasion of neutralizing antibody responses, altered entry efficiency and host protease usage[30], and notably, reduced lung pathology and disease severity in both animal models and patients[9–12]. Continued evolution within the Omicron lineage gave rise to multiple new Omicron variants, including the emergence of BA.2.86/JN.1, raising concerns about the potential reacquisition of lung pathogenicity and subsequent manifestation of severe disease. In this study, we comprehensively characterized the fitness of contemporary Omicron variants XBB.1.5, XBB.1.16, EG.5.1, and JN.1 in vivo in Syrian hamsters, as well as in vitro in primary human upper and lower respiratory tract epithelium. We demonstrate that infection with contemporary Omicron variants does not result in significant lung damage in Syrian hamsters nor in cellular damage in human alveolar organoids. Despite reports of Omicron BA.2.86 spike mediating robust entry of pseudotyped particles into a lung-derived cell line[31], we observed no virus replication of the highly related JN.1 variant in ihLOs, and significantly reduced virus replication in the lungs of JN.1-infected hamsters compared to both Delta and other Omicron variants. In contrast, contemporary Omicrons exhibit substantially increased fitness in human nasal epithelium compared to both Delta and Omicron lineage founder BA.1. This observation has been corroborated by several recent studies[32,33]. Notably, the BA.1 and JN.1 variants showed reduced virus replication in the lower and upper respiratory tract, respectively, compared to their immediate predecessors. This lower replication efficiency of JN.1 compared to its immediate predecessor EG.5.1. was observed both in our in vivo hamster model and our in vitro human models which is consistent with recent reports[5]. Both these variants are also associated with a large antigenic change compared to their predecessors, indicating a fitness cost associated with major antigenic change. Overall, our data suggest a selection pressure toward increased replication in the upper respiratory tract in the absence of lower respiratory tract pathogenesis, which is likely advantageous for transmission in the human population.

Despite the generally attenuated phenotype reported for Omicron BA.1 compared to previous variants both in vitro and in animal models, Omicron rapidly became the dominant circulating lineage. Paradoxically, airborne transmission of BA.1 in the Syrian hamster model is less efficient compared to previous variants[34]. Despite substantially increased replication in the upper respiratory tract for contemporary Omicron variants, particularly EG.5.1, we observed only a modest increase in airborne transmission efficiency of EG.5.1 compared to BA.1 (60% BA.1 to 70% EG.5.1). No airborne transmission of JN.1 was observed despite the relatively modest reduction of viral shedding in hamsters and significantly increased virus replication in human nasal

epithelium compared to BA.1. This contrasts recent reports indicating that BA.2.86 is more transmissible than EG.5.1 in the human population[5]. Together, these data highlight the importance of additional factors beyond virus replication in the upper respiratory tract to govern airborne transmission efficiency. The emergence of Omicron variants is associated with substantial changes in spike and subsequent evasion of both vaccine- and infection-elicited neutralizing antibody responses[35,36], demonstrating the sizeable impact that pre-existing immunity has had on the evolution of this lineage. Prior immunity substantially impacts inter-variant competitiveness during transmission. In immunologically naïve hamsters, Delta entirely outcompeted Omicron BA.1 in transmission chains. Detection of Omicron BA.1 transmission only occurred in vaccinated or previously infected hamsters, and this observation was linked to limited antibody neutralization cross-reactivity against BA.1[37]. Thus, the enhanced ability of Omicron variants, particularly those with substantial antigenic drift such as BA.1 and JN.1, to evade the adaptive immune response likely contributes to their increased transmissibility compared to previous variants in a human population that is no longer immunologically naïve. In this study, we compared virus-intrinsic transmission efficiency in immunologically naïve hamsters and did not observe transmission of JN.1 in contrast to BA.1. Future studies should investigate how prior immunity impacts both transmissibility and pathogenesis of JN.1.

The emergence of the Omicron lineage and subsequent evolution within this lineage has resulted in continuous antigenic drift and evasion of adaptive immune responses, particularly neutralizing antibodies. While BA.2.86 has been reported to be less immune evasive compared to XBB variants[36], it remains highly resistant to neutralizing antibodies[31,35]. The interaction between Omicron variants and the innate immune system has been less well characterized compared to adaptive immune responses. We examined differences in the host IFN response to infection with Delta, BA.1, and contemporary Omicrons to gauge the innate immune evasion potential of these variants. Omicrons, in general, elicited a stronger ISG response compared to Delta, particularly in the upper respiratory tract, suggesting a loss of innate immune evasion proficiency in this lineage consistent with previous reports[33]. In ihLOs, XBB.1.16 induced a particularly strong ISG response, suggesting a particular sensitivity to innate immune sensing for this variant. Notably, XBB.1.16-infected hamsters also displayed elevated lung histopathology scores. Together, these data suggest that XBB.1.16 triggers a stronger immune response, which may contribute to the increased pathology observed in the lungs, but may also help control virus replication. The lack of ISG response observed in ihLOs inoculated with other Omicron variants is likely due to the absence of replication. We have previously reported that an IFN response to Omicron BA.1 infection in lung organoids was only observed if there was detectable virus replication[13]. This aligns with the overall increased replication and selection towards the upper respiratory tract of Omicron variants[33]. Differences in the ISG-induction profile between the Omicron variants further suggest differential abilities among the Omicron variants to both trigger and counteract the innate immune response[38], indicative of continued evolution and adaptation within the Omicron lineage. Further studies are warranted to investigate how Omicron variants overcome this ISG response to replicate efficiently in the upper respiratory tract.

This study has several limitations. While human primary epithelial cultures can robustly model virus-host interactions, they lack aspects of cellular diversity and immune memory that contribute to pathogenesis at the organ level. Therefore, the lack of cytotoxicity observed in infected nasal ALI and lung organoid cultures (ihLO) may not fully represent pathogenesis in vivo. Furthermore, the in vivo experiments in hamsters do not consider pre-existing SARS-CoV-2 immunity, which is widely present in the human population and can substantially impact both transmission and pathogenesis. Therefore, the reduced

replication capacity and transmissibility of JN.1 in the immunologically naïve hamster model might not fully capitulate the growth advantage JN.1 has in the human population with pre-existing immune pressures.

This study integrates data generated from in vivo animal models and in vitro primary human epithelial culture models to comprehensively characterize Omicron variants in the upper and lower respiratory tract. In both model systems, we observed robust fitness of contemporary Omicron variants in the upper respiratory tract along with reduced replication and pathology in the lung, compared to previous VOCs. Together, these data support an evolutionary trend in the Omicron lineage toward adaptation to the upper respiratory tract.

## Methods

### Ethics statement

Animal experiments were conducted in an AAALAC International-accredited facility and were approved by the Rocky Mountain Laboratories Institutional Care and Use Committee following the guidelines put forth in the Guide for the Care and Use of Laboratory Animals 8th edition, the Animal Welfare Act, United States Department of Agriculture and the United States Public Health Service Policy on the Humane Care and Use of Laboratory Animals. The Institutional Biosafety Committee (IBC) approved work with SARS-CoV-2 at biosafety level 3 and subsequent inactivation of samples for removal from high containment[39].

### Cells and viruses

Cell lines were maintained in Dulbecco's modified Eagle's medium (DMEM) supplemented with 10% fetal bovine serum (FBS), 1mM L-glutamine, 50 μ/mL penicillin, and 50μg/mL streptomycin unless mentioned otherwise. A549 cells expressing human ACE2, hamster ACE2, and human TMPRSS2 were generated as described before[40,41]. HEK293T and VeroE6 cells were obtained from ATCC, Vero E6-TMPRSS2-T2A-ACE2 cells from BEI resources and all cells were propagated from laboratory stocks. Nasal MucilAir ALI cultures were purchased from Epithelix and maintained using the MucilAir culture medium according to the manufacturer's instructions.

SARS-CoV-2 virus stocks were propagated on VeroE6 cells in DMEM supplemented with 2% FBS. The following SARS-CoV-2 variants were used, and indicated SNPs compared to the patient sample sequence were detected during propagation.: **Delta** (SARS-CoV-2/USA/CA-VRLC086/2021; EPI_ISL_2987140) provided by Andrew Pekosz with following SNPs detected: NSP3 P1228L 100%; NSP14 A394V 100% (both observed in delta variants), **BA.1** (hCoV-19/USA/GA-EHC-2811C/2021; EPI_ISL_7171744) provided by Mehul Suthar, **XBB.1.5** (SARS-CoV-2/USA/HP40900/2022; EPI_ISL_16026423) provided by Andrew Pekosz with following SNPs detected: NSP2 G339S, 74%; NSP6 M47T, 61%; E V14 (INDEL), 74%; orf7a A50A 58%, **XBB.1.16** (hCoV-19/USA/MD-HP46342-PIDFPPISFA/2023; EPI_ISL_17394984) provided by Andrew Pekosz, **EG.5.1** (hCoV-19/USA/CA-Stanford-147_S01/2023; EPI_ISL_17977757) provided by Merideth Gardner, and **JN.1** (hCoV-19/USA/NY/PV96109/2023; EPI_ISL_18563626) provided by CDC. Comparison of TCID50 and RNA copies per virus stock: Delta 1.26E + 07 TCID50/mL vs 2.62E + 08 copies/mL. BA.1 1.00E + 06 TCID50/mL vs 9.83E + 07 copies/mL. XBB.1.5 3.16E + 06 TCID50/mL vs 8.99E + 06 copies/mL. XBB.1.16 1.00E + 07 TCID50/mL vs 9.62E + 07 copies/mL. EG.5.1 3.98E + 06 TCID50/mL vs 6.84E + 07 copies/mL. JN.1 3.98E + 05 TCID50/mL vs 3.99E + 08 copies/mL.

### ihLO generation, culture, and infection

Episomal hiPSCs (Gibco A18945) were maintained according to the manufacturer's instructions. Differentiation of iPSCs into ihLOs was conducted according to established methodology[27]. Briefly, iPSCs were first differentiated into definitive endoderm, then anterior foregut endoderm, and lastly into lung progenitor cells over the course of 14 days. Efficient induction of definitive endoderm was confirmed by flow cytometry analysis of c-Kit and CXCR4 co-expression using a FACS Symphony A5 instrument (BD Biosciences). NKX2.1 + distal lung progenitor cells were isolated via fluorescently-activated cell sorting (FACS) using a MACSQuant Tyto Cell Sorter (Miltenyi) based on surface marker expression: $CD47^{HI}$, $CD26^{LO}$. Expression of NKX2.1 in $CD47^{HI}$ and $CD26^{LO}$ cells was confirmed by intracellular staining for NKX2.1 analyzed via flow cytometry. Sorted distal lung progenitor cells were seeded for three-dimensional culture at 200 cells/μL in 10 μL droplets of 66% vol/vol growth factor reduced, LDEV-free, phenol red-free Matrigel (Corning) supplemented with 33% vol/vol CK-DCI culture medium[27]. Cells were incubated at 37 °C for 30 min to allow Matrigel droplets to solidify, after which CK-DCI culture medium was overlaid. Distal lung progenitor cells were cultured for 10 days to allow the formation of ihLOs. At this stage, ihLO cells were sorted by FACS to enrich AT2 cells based on the expression of CPM. ihLOs were cultured from sorted $CPM^{HI}$ cells as described above. ihLOs were passaged via dissociation into single cells with TrypLE (Thermo Fisher Scientific), as described previously[13]. Antibodies used were as follows: anti-c-kit-PE (clone 104D2, BioLegend), anti-CXCR4-Brilliant Violet 605 (clone 12G5, BioLegend), anti-CD47-Brilliant Violet 421 (clone CC2C6, BioLegend), anti-CD26-APC (clone BA5b, BioLegend), anti-NKX2.1 (clone EP1584Y, Abcam), CPM (clone WK, FUJIFILM Wako Chemicals), anti-rabbit IgG-PE (12-4739-81, Thermo Fisher Scientific), anti-mouse IgG-PE (12-4010-82 Thermo Fisher Scientific).

To infect ihLOs with SARS-CoV-2, organoids were dislodged from Matrigel droplets via pipetting in PBS using a wide orifice 1000 μL pipet tip. A fraction of organoids were dissociated into a single-cell suspension and counted to facilitate the calculation of MOI. Organoids were incubated with the virus at MOI 0.1 in CK-DCI medium supplemented with 10 μM Y-27632 (CK-DCI-Y) for 1 h at 37 °C with rotation. The inoculum was removed, and organoids were washed once in PBS and plated at 2500 cells per μL in 10 μL Matrigel droplets as described above. After incubation at 37 °C for 30 min, 600 μL of CK-DCI-Y medium was overlaid.

### Electron microscopy

ihLOs were fixed with 2% paraformaldehyde/ 2.5% glutaraldehyde in 0.1 M Sorensen's buffer overnight, followed by processing for transmission electron microscopy. 0.01% malachite green was added to the fixative to help visualize ihLOs. Briefly, ihLOs were washed with 0.1 M sodium cacodylate buffer and embedded in low-melt agarose. ihLOs were post-fixed with 0.5% $OsO_4$/0.8% $K_4Fe(CN)_6$ in 0.1 M sodium cacodylate, washed with 0.1 M sodium cacodylate buffer, and stained with 1% tannic acid. After water washes, ihLOs were further osmicated with 2% $OsO_4$ in 0.1 M sodium cacodylate buffer, washed with buffer, and stained with 1% uranyl acetate overnight at 4 °C. ihLOs were washed with $dH_2O$, dehydrated in a graded ethanol series, infiltrated with Spurr's resin, and polymerized overnight in a 65 °C oven. Thin sections were cut using a Leica EM UC6 ultramicrotome (Leica Microsystems) and viewed using a Hitachi HT7800 transmission electron microscope at 80 kV (Hitachi High-Tech America). Images were captured with an AMT digital camera system (Advanced Microscopy Techniques).

### Infection of nasal ALI cultures

Nasal ALI cultures were infected with SARS-CoV-2 variants at MOI 0.1. 150 μL virus inoculum was added to the apical chamber and incubated for 1 h at 37 °C. The inoculum was removed, and cells were washed twice with PBS. The basolateral medium was removed and replaced with 500 μL fresh MucilAir culture medium (Epithelix) at each sampling timepoint. For apical sampling, apical chambers were washed twice with 250 μL of culture medium, and wash samples were combined.

## Trans-epithelial electrical resistance (TEER) measurement

After collection of basolateral media and apical washes, 300 μL or 500 μL of PBS was added to the apical and basolateral chambers, respectively. TEER was measured using an EVOM3 voltmeter (World Precision Instruments). TEER measurement from the blank control well was subtracted before the calculation of Ohms/cm$^2$.

## Cytotoxicity assay

LDH release into culture supernatant was measured using the LDH-Glo Cytotoxicity Assay (Promega) according to the manufacturer's instructions. Provided LDH standards were run in parallel to calculate LDH concentration.

## Pseudotype entry assay

Vesicular Stomatitis Virus (VSV) particles were pseudotyped with heterologous SARS-CoV-2 spike proteins of indicated variants. The spike sequences were appended with a 5' kozak sequence, and the 3' end of the spike sequence was truncated by 19 aa and appended with a tetra-glycine linker followed by a FLAG-tag sequence (DYKDDDDK). Modified spike sequences were synthesized and inserted into pcDNA3.1(+) (GenScript). Pseudovirus particle production was performed as previously described[42]. Briefly, HEK293T cells were seeded in 6-well plates and transfected with 2 μg spike expressing plasmid, VSV-G or GFP (negative control). 24 h post-transfection, cells were inoculated with seed particles (VSVΔG-luc/GFP + VSV-G) at MOI3 in DMEM supplemented with 2% FBS. After 1 h incubation at 37 °C, cells were washed three times with PBS, and fresh DMEM supplemented with 2% FBS was added. 48 h post-inoculation, culture supernatants were harvested, cleared from debris by centrifugation (500 × g, 5 min), aliquoted, and stored at − 80 °C until use.

To assess the entry efficiency of the different pseudovirus particles carrying SARS-CoV-2 spikes, the infectious titer of the generated pseudovirus stocks was determined by serial dilution. For this, indicated target cells were infected with a titrated challenge of serially diluted virus. 24 hours post-inoculation, cells were fixed, and the number of GFP-positive cells was determined using flow cytometry (BD FACS Symphony). Shown values are the mean of triplicate ($n = 3$) estimations of the titer extrapolated from different doses within the linear range and background subtracted from GFP pseudotyped particles. A typical result from at least two independent experiments is shown.

## Western Blot

ACE2 and TMPRSS2 expression was analyzed in cell lysates by western blot analysis. To this end, cells were seeded the day before harvest at 10^6 cells/well into a 6-well plate. Cells were washed once with PBS, harvested in SDS lysis buffer (12.5% glycerol, 175 mM Tris-HCl (pH 8.5), 2.5% SDS, 70 mM 2-mercaptoethanol, and 0.5% bromophenol blue), heated at 70 °C for 10 min and sonicated. To assess the incorporation of SARS-CoV-2 spike proteins into pseudovirus particles, the pseudovirus particles were concentrated through a sucrose cushion (20% w/v sucrose in PBS) by high-speed centrifugation (20,000 × g, 2 h, 4 °C). Concentrated particles were harvested in SDS lysis buffer and heated at 70 °C for 10 min.

Protein lysates were separated on NuPage 4–12% Bis-Tris polyacrylamide gels (Invitrogen) and transferred onto nitrocellulose membranes. Blots were probed with the following primary antibodies against ACE2 (21115-1-AP, Proteintech), TMPRSS2 (ab109131, Abcam), GAPDH (60004-1-Ig, Proteintech), SARS-CoV-2 spike S2 (MAB10557, R&D systems), VSV-M[23H12] (Ab01404-2.0, Absolute Antibody). Subsequently, membranes were probed with species-specific fluorescently labeled secondary antibodies goat anti-rabbit IgG (SA5-10036, Thermo Fisher Scientific) or goat anti-mouse IgG (35519 or SA5-10176, Thermo Fisher Scientific) and scanned using a LiCor Odyssey scanner.

## Inoculation of Syrian hamsters

Four-to-six-week-old female and male Syrian hamsters (Envigo) were randomly assigned to experimental groups. Before infection, baseline body weights were measured. Under isoflurane anesthesia, hamsters were inoculated with $1 \times 10^4$ TCID$_{50}$ of SARS-CoV-2 XBB.1.5, XBB.1.16, EG.5.1, and JN.1 variants in a volume of 40 μl (intranasal) or 100 μl (intratracheal) diluted in DMEM. Intranasally inoculated hamsters ($n = 4$ per virus) were swabbed and weighed daily until 5 DPI; animals were euthanized at 5 DPI and tissues were collected. For the intratracheal challenge, the hamsters were restrained on a hang board, and using tissue-friendly forceps the tongue was retracted. After visualizing the trachea with a speculum and light source, a 1 mL syringe with a 21–25 gauge flexible catheter tip was used to inoculate the hamsters. Six hamsters were swabbed and weighed on days 1, 3, 5, and 7 post-inoculation, and serum was collected on day 28. On 3, 5, and 7 DPI, 4 hamsters per virus were euthanized, and tissues were collected.

## Transmission experiment

Four-to-six-week-old male Syrian hamsters (Envigo) were randomly assigned to experimental groups. Airborne and contact transmission was examined by co-housing hamsters (1:1:1) in specially designed cages with a 3D-printed perforated plastic divider dividing the living space in half[21]. This divider prevented direct contact between the donor/primary infected and the airborne sentinel hamster and the movement of bedding material. The contact sentinels were housed on the same side as the donor animal and the airborne sentinels were placed in the direction of the airflow. Regular bedding was replaced by alpha-dri bedding to avoid the generation of dust particles. Donor hamsters were inoculated intranasally as described above with either BA.1, EG.5.1 or JN.1 ($N = 10$), single housed and on 1 DPI (36 h post-inoculation) re-housed to the transmission cages. Sentinel hamsters were then placed in the transmission cages and exposure lasted for 24 h. Experiments were performed with cages placed into a standard rodent cage rack, under normal airflow conditions. Donor animals were swabbed orally on 1 DPI and 2 DPI, then euthanized. Sentinel animals were swabbed on 1-day post-start of exposure (DPE) and 3 DPE. Serum was collected on 14 DPE.

## SARS-CoV-2 spike enzyme-linked immunosorbent assay (ELISA)

Seroconversion of hamsters was determined via enzyme-linked immunosorbent assay (ELISA). In short, Maxisorp plates (Nunc) were coated with 50 ng Lineage A spike recombinant protein (Sino biological CAT# 40589-V08B1) and incubated overnight at 4 °C. Plates were blocked with Casein in PBS (ThermoFisher) for 1 h at room temperature (RT) followed by addition of 1:100 diluted serum samples in blocking buffer for 1 h at RT. Spike-specific antibodies were detected with a secondary goat anti-hamster IgG antibody (seracare CAT# 5220-0371) for 1 h at RT. After washing with PBS-T (0.1%Tween), plates were visualized with a KPL TMB 2-component peroxidase substrate kit and stop solution (SeraCare CAT# 5120-0047). Plates were read at 450 nm using a multi-mode plate reader (BioTek). Sample positivity was calculated based on the threshold of the average plus 3x the standard deviation of the negative control hamster serum.

## Virus titration

Virus titer was determined via TCID50 assay with ten-fold serial dilutions of the sample. After six days, cells were read for cytopathic effect (CPE), and the titer was calculated using the Spearman-Karber method. Virus stocks were titrated on VeroE6 cells using ten-fold serial dilutions of virus stock. For hamster experiments, tissue samples were weighed and subsequently homogenized in 1 mL DMEM supplemented with 2% FBS (DMEM-2). Ten-fold serial dilutions of the homogenate were used to inoculate VeroE6 cells, and the virus titer was calculated as above. For ihLO experiments, virus titration was conducted using Vero E6-

TMPRSS2-T2A-ACE2 cells to increase sensitivity. Cells were inoculated with ten-fold serial dilutions of cell culture supernatant. Cells were centrifuged at $300 \times g$ for 30 min, followed by a 30 min incubation at 37 °C and 5% $CO_2$. Cells were washed once with PBS, and the medium was replaced with DMEM-2. Virus titer was calculated as described above.

### SARS-CoV-2 RNA detection

For animal swabs, 140 μl swab media was used for viral RNA extraction using the QIAmp Viral RNA Kit (Qiagen) using the automated QIAcube HT system (Qiagen) according to the manufacturer's instructions. For animal tissues, RNA was manually isolated using the RNeasy Mini kit (Qiagen) according to the manufacturer's instructions. SARS-CoV-2 subgenomic and genomic viral RNA were detected using qRT-PCR[43]. Briefly, isolated RNA was run using a TaqMan Fast Virus One-Step Master Mix (Applied Biosystems) on a Quant-Studio 3 Flex Real-Time PCR system (Applied Biosystems). RNA standards with known copy numbers were used to generate a standard curve and calculate sample copy numbers. For nasal ALI and ihLO experiments, RNA was extracted from 140 μl of apical or basolateral culture using the QIAamp Viral RNA Mini kit (Qiagen), or from cells lysed in Buffer RLT using the RNeasy Mini Kit (Qiagen), according to the manufacturer's instructions. SARS-CoV-2 gRNA[43] and sgRNA[44] were quantitated using one-step qRT-PCR with 2 μl of RNA input conducted using the QuantiNova Probe RT-PCR kit (Qiagen) according to the manufacturer's instructions and run on a QuantStudio 6 instrument (Thermo Fisher Scientific). Serial dilutions of RNA standards with known copy numbers were run in parallel in each run to calculate RNA copy numbers in the standard.

### Quantification of IFN-stimulated gene expression

RNA was extracted from cell lysate as described above. cDNA was generated using the SuperScript IV VILO kit with EZDNase (Thermo Fisher Scientific) according to the manufacturer's instructions. qPCR was performed using the TaqMan Fast Advanced Master Mix (ThermoFisher) on a QuantStudio 6 instrument (Thermo Fisher Scientific). ACTB was used as an endogenous control for relative quantification analysis, and data were normalized to mock-infected timepoint-matched controls. Probes were labeled with FAM/ZEN/IowaBlackFQ (Integrated DNA Technologies). Primer/probe sequences are provided in the Supplementary Table 3.

### Virus neutralization

Hamster sera were heat-inactivated at 56 °C for 30 min. Two-fold serial dilutions of the sera were prepared starting at a 1:50 dilution in DMEM supplemented with 2% FBS. 150 $TCID_{50}$ of the respective SARS-CoV-2 variants were added to the diluted sera and incubated for 1 h at 37 °C and 5% CO2. After incubation, the virus-serum mixture was added to VeroE6 cells, and the cells were incubated for an additional six days. CPE was assessed on day 6 post-inoculation and the virus neutralization titer (ND100%) expressed as the reciprocal value of the highest dilution of serum that still inhibited virus replication was determined. Three different positive serum controls were included.

### Antigenic cartography

Antigenic maps were constructed as described previously[45,46] using antigenic cartography software from https://acmacs-web.antigenic-cartography.org. Briefly, antigenic mapping uses multidimensional scaling to position antigens (viruses) and sera on a map that represents their antigenic relationship. The positions on the map of both antigens and sera were optimized to minimize the error between the target distance set by the observed pairwise virus-serum combinations in the virus neutralization assay described above. Antigenic maps were constructed in 2 dimensions.

### Histopathology

Tissues were fixed for a minimum of 7 days in 10% neutral buffered formalin. Tissues were processed with a Sakura VIP-6 Tissue Tek, on a 12 h automated schedule, using a graded series of ethanol, xylene, and PureAffin. Prior to staining, embedded tissues were sectioned at 5 μm and dried overnight at 42 °C. Sections were stained with hematoxylin and eosin stain (H&E) for evaluation of histologic lesions. To detect SARS-CoV-2 antigen by immunohistochemistry (IHC), tissues were labeled with anti-NP-1 antibody (GenScript U864YFA140-4/CB2093 NP-1). Briefly, tissues were labeled with a 1:1000 dilution of antibody, followed by incubation with a secondary horseradish peroxidase linked anti-Rabbit antibody (Vector Laboratories ImPress VR anti-rabbit IgG polymer (# MP-6401)) and counterstaining with hematoxylin. The IHC assay was carried out on a Discovery ULTRA automated-staining instrument (Roche Tissue Diagnostics) with a Discovery ChromoMap DAB kit (Ventana Medical Systems, #760-159). A board-certified veterinary anatomic pathologist evaluated all tissue slides. Histological lesions from lung sections stained with H&E were categorized into bronchiolitis, interstitial pneumonia, cellular exudate, type II pneumocyte hyperplasia, edema, and vasculitis. For each category, histological lesions were scored on a scale of none = 0, rare = 1, mild = 2, moderate = 3, and severe = 4.

### Statistical analysis

Animals were randomly assigned to the experimental groups. For in vivo studies, data distribution was assumed to be non-normal, and non-parametric tests were applied where appropriate. No data or animals were excluded from the analysis. Significance tests were performed as indicated where appropriate using Prism 9 (GraphPad Software). $P$-values < 0.05 are shown.

### Reporting summary

Further information on research design is available in the Nature Portfolio Reporting Summary linked to this article.

## Data availability

Data included in this manuscript are provided in the Source Data file and have been deposited in Figshare at: https://doi.org/10.6084/m9.figshare.26035741. Source data are provided in this paper.

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

## Acknowledgements

We would like to thank J. Prado-Smith and D. Long for assistance with histology and immunohistochemistry of hamster tissues, S. Antonioli-Schmit for assistance with electron microscopy sample preparation and imaging, and Alexander Stewart and SJ Tudor for graphic arts assistance. This study was supported by the Intramural Research Program of NIAID, NIH.

## Author contributions

Conceptualization: A.W., M.F., E.d.W., and V.J.M. Methodology: A.W., M.F., J.R.P., C.K.Y., B.N.W., E.d.W., and V.J.M. Investigation: A.W., M.F., J.R.P., C.K.Y., K.G., S.G., J.E.S., T.L., B.N.W., F.K., R.K.M., S.v.T., B.S., N.v.D., C.A.R., E.d.W., and V.J.M. Visualization: A.W., M.F., and K.G. Supervision: E.d.W and V.J.M. Writing - original draft: A.W., M.F., E.d.W., and V.J.M. Writing - review & editing: all authors.

## Funding

## Competing interests

The authors declare no competing interests.
