## [Peer Review file · Nature Communications]

Evolution of Omicron lineage towards increased fitness in the upper respiratory tract in the absence of severe lung pathology

Corresponding Author: Dr Arthur Wickenhagen

Version 0:

Reviewer comments:

Reviewer #1

(Remarks to the Author)

Wickenhagen and colleagues investigated entry, spread and pathogenesis of XBB variants, EG5.1 and JN.1 in cell line models, primary cells and a hamster model. In brief, they report that variants post BA.1 exhibit increased capacity to enter ACE2 transfected cell lines. In the hamster model, XBB viruses and EG5.1 replicated and caused signs of disease to comparable levels while JN.1 was attenuated. This trend remained but more robust signs of disease were observed upon intratracheal application of the virus (as compared to intranasal). Further, the Omicron variants were shown to replicate to higher levels than Delta in cultured human nasal epithelium but largely failed to replicate in cultured lung epithelium. Finally, XBB variants and EG5.1 but not JN.1 were efficiently transmitted between hamsters. This is a comprehensive study and of some interest to the field. However, some points remain open.

Major

Based on figure 1A no firm statement regarding entry efficiency/ACE2 affinity is possible since marked differences in spike incorporation into particles were observed (or differential reactivity with the S2 antibody, which would require usage of tagged S proteins). In order to allow for robust conclusions, pseudotypes harboring roughly comparable levels of spike (entry efficiency) and recombinant proteins (ACE2 affinity) are needed. In figure S1, marker bands as well as S0 (if observed) and S2 bands should be indicated.

The information on the impact of infection on ISG expression is of limited value since only 3 ISGs were studied and induction apparently correlated with infection efficiency. It should also be examined whether directed expression of the ISGs in cell lines and pre-treatment of nasal epithelium with IFN differentially impacts infection by the variants studied.

In the hamster model, JN.1 was found to be attenuated and not transmissible, despite robust replication in hamster nasal epithelium in vivo and human nasal epithelial cells in vitro. The lack of transmission is in sharp contrast with the efficient spread of JN.1 in the human population. This points towards significant limitations of the hamster model and/or problems with the JN.1 isolate used and should be discussed in more detail and/or subjected to further investigation. For instance, did JN.1 exhibit altered aerosol stability and were mutations observed that might hint towards adaptation to the rodent host.

Minor

It is unclear which controls are shown in figure 2C.

Reviewer #2

(Remarks to the Author)

Wickenhagen et al. characterized the properties of WA-1, Delta, Omicron BA.1, XBB.1.5/1.16, EG5.1 and JN.1 variants. The authors compared replication of these variants and induction of IFN responses in human primary epithelial cells models.

They also used the hamster model to compare their pathology and their transmission, after direct contact between infected animals or through airborne transmission. They also established an antigenic mapping of the different variants. Overall, the results are interesting, providing a detailed overview of the different variants including the recent JN.1 strain, and the selective advantage. The paper is well written.

Major points:

1. The authors titrated the viral stocks on Vero E6 for the hamster experiments and on Vero E6 ACE2+/TMPRSS2+ for primary epithelial experiments. Why were two different cell lines used?
2. It has been shown that Omicron variants exhibit reduced ability to replicate in different cells such as Vero E6 or Calu-3 (Planas et al., *Nature Com*, 2024; Tamura et al., *Cell Host Microbe*, 2024). The authors used Vero E6 to determine viral titers, which may bias the calculation of the MOI. For the hamster experiments, a figure showing the amount of inoculated RNA for each variant is needed.
3. Fig 5: The quantification of infectious virus does not seem to match the subgenomic RNA data (Fig S5). Genomic RNA should be quantified as in Figure 3.
4. Fig 6: The transmission experiments do not reflect the human situation, with JN.1 being more efficiently transmitted than previous variants. Does this suggest that the in vivo hamster model is not a good model for studying the viral fitness and pathogenicity of JN.1? The role of previous immunity should be further discussed.
5. Fig 7: The neutralization data (ED50) could be included in an additional figure. Neutralizations were performed using Vero E6 cells using cytopathic effect as a readout readout for neutralizing activity of sera. As mentioned in point 1, Omicron variants, poorly replicate in these cells and are less cytopathic compared to previous variants. This presents a major concern for comparing neutralization activity against Omicron variants in Vero E6 cells. this could be further discussed as a limitation of the study.
6. Several groups compared virological and immunological properties of Omicron variants including BA.2.86/JN.1. Several references need to be added and commented in the discussion:

Tamura T, Mizuma K, Nasser H, Deguchi S, Padilla-Blanco M, Oda Y, Uriu K, Tolentino JEM, Tsujino S, Suzuki R, Kojima I, Nao N, Shimizu R, Wang L, Tsuda M, Jonathan M, Kosugi Y, Guo Z, Hinay AA Jr, Putri O, Kim Y, Tanaka YL, Asakura H, Nagashima M, Sadamasu K, Yoshimura K; Genotype to Phenotype Japan (G2P-Japan) Consortium; Saito A, Ito J, Irie T, Tanaka S, Zahradnik J, Ikeda T, Takayama K, Matsuno K, Fukuhara T, Sato K. Virological characteristics of the SARS-CoV-2 BA.2.86 variant. *Cell Host Microbe*. 2024 Feb 14;32(2):170-180.e12.

Planas D, Staropoli I, Michel V, Lemoine F, Donati F, Prot M, Porrot F, Guivel-Benhassine F, Jeyarajah B, Brisebarre A, Dehan O, Avon L, Bolland WH, Hubert M, Buchrieser J, Vanhoucke T, Rosenbaum P, Veyer D, Péré H, Lina B, Trouillet-Assant S, Hocqueloux L, Prazuck T, Simon-Loriere E, Schwartz O. Distinct evolution of SARS-CoV-2 Omicron XBB and BA.2.86/JN.1 lineages combining increased fitness and antibody evasion. *Nat Commun*. 2024 Mar 13;15(1):2254.

Li P, Faraone JN, Hsu CC, Chamblee M, Zheng YM, Carlin C, Bednash JS, Horowitz JC, Mallampalli RK, Saif LJ, Oltz EM, Jones D, Li J, Gumina RJ, Xu K, Liu SL. Neutralization escape, infectivity, and membrane fusion of JN.1-derived SARS-CoV-2 SLip, FLiRT, and KP.2 variants. *Cell Rep*. 2024 Jul 17;43(8):114520.

Qu P, Xu K, Faraone JN, Goodarzi N, Zheng YM, Carlin C, Bednash JS, Horowitz JC, Mallampalli RK, Saif LJ, Oltz EM, Jones D, Gumina RJ, Liu SL. Immune evasion, infectivity, and fusogenicity of SARS-CoV-2 BA.2.86 and FLip variants. *Cell*. 2024 Feb 1;187(3):585-595.e6.

Zhang L, Kempf A, Nehlmeier I, Cossmann A, Richter A, Bdeir N, Graichen L, Moldenhauer AS, Dopfer-Jablonka A, Stankov MV, Simon-Loriere E, Schulz SR, Jäck HM, Čičin-Šain L, Behrens GMN, Drosten C, Hoffmann M, Pöhlmann S. SARS-CoV-2 BA.2.86 enters lung cells and evades neutralizing antibodies with high efficiency. *Cell*. 2024 Feb 1;187(3):596-608.e17

7. Fig 1: Statistics are missing. The figure is difficult to read with all the bars.

Reviewer #3

(Remarks to the Author)

The manuscript of Wickenhagen et al. adds important aspects to our understanding of SARS-CoV-2 Omicron evolution. The authors answer a relevant scientific question with a combination of in-vivo experiments in the Syrian hamster and in-vitro experiments in primary human cells of the respiratory tract. Although these experiments are interesting and valuable for the evaluation of the specific characteristics of the evaluated Omicron variants, an overall statement of SARS-CoV-2 evolution would need more controls, which are lacking in specific experiments.

Specific:

l.53: Specify "non-naïve". Assuming you refer to immunologically naïve?

ll.61: What is meant with "Species susceptibility in the Syrian golden hamster"? Isn't it just susceptibility in the Syrian golden hamster?

ll.73 and Figure 1a: Can you explain the rationale for including the hamster ACE-2, human TMPRSS2 transfected cells? Have these experiments also been performed with hamster only (either hamster ACE-2, hamster TMPRSS2 or hamster ACE-2 only)? Otherwise I wonder how conclusive those experiments are in terms of species specificity. Alternatively please make clear why this selection was made. Furthermore, why is WA.1 used here, whereas in majority of following in-vitro experiments the authors use Delta as pre-Omicron control?

ll.82: Although there is an increase for Omicron EG.5, later variants dip again in binding. It seems there is no clear dynamic with the emergence of variants over time. Any explanations or thoughts how that mechanism then influences evolution?

ll.85: I agree that incorporation of spike is overall successful, yet variable. Can that have an influence on this experiment?

ll.97: Authors state there is no weight loss, but what about animals inoculated with EG.5 at 2 dpi? An equivalent loss of body weight is interpreted as body weight loss in the following experiment (intratracheal inoculation) of the authors.

Overall for the experiment of intranasal inoculation of hamsters I am missing a pre-Omicron control group to assess if there is indeed the evolutionary trend towards less LRT disease and replication and especially more efficient replication in the URT. Maybe historical controls of previous experiments can be incorporated or the interpretation of data should be tuned down (focus on intra-Omicron description, not general SARS-CoV-2 evolution).

ll.143: This was also briefly mentioned in your discussion, but this sentence is an overinterpretation to me. Infection of ALI cultures with SARS-CoV-2, independent of which variant chosen, barely induces any cpe. This stands in sharp contrast to observations in the field and in-vivo. Therefore the absence of cpe should not be used as argument for loss of LRT pathology. Especially since the same absence is observed for Delta.

ll.146: It would be important to hear the rationale of the authors to conduct this inoculation route. What is the relevance and how can results be translated to observations in the field? What is the added value of this elaborate in-vivo experiment?

ll.180: For the reading flow, it would be better if the last paragraph would follow upon ll. 168.

ll.195: Interesting to see that XBB.1.16 infected hLOs produce highest amount of infectious virus. In hamsters you see most of pathology here, yet not the highest amount of infectious virus (or RNA). Would be interesting to elaborate on those observations in more detail in the discussion. What can cause these differences?

ll.207: Can you visualize the comparison of LRT vs URT? Since this is a very relevant observation/question, it would be helpful to have a comprehensive figure indicating comparative findings.

ll.210: This is a very strong statement, which is not underlined with the needed data since comparisons rely on 1 pre-Omicron variant only.

ll.218: What is the rationale to include this specific Omicron variants? Especially since BA.1 was not focus of previous experiments?

ll.241: Why now Alpha and WA.1? Generally the inclusion of different variants seems to be inconsistent throughout the different experiments.

ll.297: Is this than an inherent feature of the virus or just linked to virus titers (excl. Delta)?

ll.307: Not only the transmission experiments do not consider pre-existing immunity. Also pathogenesis in-vivo experiments and infections of primary cells does not.

Figure 1: Although the authors show that Omicron is less dependent on TMPRSS2, can the human TMPRSS2 outweigh the less efficient entry via hamster ACE-2? Have statistics been performed here?

Figure 2: Results match with observations of experiments visualized in figure 1 for results in the lungs, not so much for the nose. Any ideas why?

Figure 4: How do the authors explain the significant differences observed in IHC, but not in viral titration (4D vs 4I) at 3 dpi?

Version 1:

Reviewer comments:

Reviewer #1

(Remarks to the Author)

The authors have adequately addressed the questions of this reviewer and the findings reported are of interest to the field.

Minor: "In this study we compared virus-intrinsic transmission efficiency in immunologically naïve hamsters and observed reduced transmissibility of JN.1 compared to BA.1. Future studies should investigate how prior immunity impacts both transmissibility and pathogenesis of JN.1". This statement is not fully clear to this reviewer. JN.1 transmissibility was not only reduced, it was absent. Is a different situation expected in immunologically non-naïve animals?

Reviewer #2

(Remarks to the Author)

the authors have addressed my concerns

Reviewer #3

(Remarks to the Author)

The authors replied in a satisfactory way to all raised comments and included relevant information and changes into the main text/figures.

Therefore, I don't have any further comments.

REVIEWER COMMENTS

Dear Editor,

We thank the reviewers for their constructive comments and suggestions, which have increased the strength and thoroughness of our work. We have replied to the comments below and in our revised manuscript. To address these comments, we have added 10 new figure panels, one new supplementary figure, and three new supplementary tables. We hope you agree that our manuscript is now suitable for publication in Nature Communications.

Sincerely, also on behalf of the co-authors,

Vincent J Munster, PhD

Reviewer #1 (Remarks to the Author):

Wickenhagen and colleagues investigated entry, spread and pathogenesis of XBB variants, EG5.1 and JN.1 in cell line models, primary cells and a hamster model. In brief, they report that variants post BA.1 exhibit increased capacity to enter ACE2 transfected cell lines. In the hamster model, XBB viruses and EG5.1 replicated and caused signs of disease to comparable levels while JN.1 was attenuated. This trend remained but more robust signs of disease were observed upon intratracheal application of the virus (as compared to intranasal). Further, the Omicron variants were shown to replicate to higher levels than Delta in cultured human nasal epithelium but largely failed to replicate in cultured lung epithelium. Finally, XBB variants and EG5.1 but not JN.1 were efficiently transmitted between hamsters. This is a comprehensive study and of some interest to the field. However, some points remain open.

Major

Based on figure 1A no firm statement regarding entry efficiency/ACE2 affinity is possible since marked differences in spike incorporation into particles were observed (or differential reactivity with the S2 antibody, which would require usage of tagged S proteins). In order to allow for robust conclusions, pseudotypes harboring roughly comparable levels of spike (entry efficiency) and recombinant proteins (ACE2 affinity) are needed. In figure S1, marker bands as well as S0 (if observed) and S2 bands should be indicated.

We agree with the reviewer that there are differences in the incorporation of the different spikes. Our goal was not to compare ACE2 affinity or entry between the different variants directly, we only wanted to compare within each pseudotype (intra-pseudotype comparison). Each pseudotype is titrated on cells expressing ACE2, ACE2 and TMPRSS2 or TMPRSS2 and hamster ACE2. To better highlight this comparison, we have moved the initial entry data into the supplement and in the main figure only show the comparison between TMPRSS2 usage and the differential usage of hamster ACE2 over human ACE2 for each pseudotype. We have modified the text to highlight within-pseudotype comparisons +/- TMPRSS2, or human vs hamster ACE2 (lines 73-93). Additionally, we have included the raw

WB images in the supplementary figure 1 together with markers for size and S0, S2 as observed.

The information on the impact of infection on ISG expression is of limited value since only 3 ISGs were studied and induction apparently correlated with infection efficiency. It should also be examined whether directed expression of the ISGs in cell lines and pre-treatment of nasal epithelium with IFN differentially impacts infection by the variants studied.

The reviewer is correct that we only assayed 3 ISGs. We have now increased the number of ISGs assayed to a total of 7 (Figure 7), and an additional 4 cytokines measured in Nasal ALI samples (Figure S6). Induction of these ISGs was similar to those previously assayed and supports our conclusion that infection with Omicron variants results in a stronger ISG response compared to infection with Delta. Further, we would like to highlight that ISG induction is not solely correlated with infection efficiency: Delta replicates productively in lung organoids (Figure 5A) but does not induce an ISG response (Figure 7B-H). We have clarified this in the text (lines 269, 273). Although the reviewer is correct that it would be interesting to test the effect of IFN pretreatment, this experiment has limited relevance for our conclusion that more recent Omicron variants exhibit enhanced replicative fitness in the upper respiratory tract despite higher induction of an ISG response. Therefore, this experiment was not feasible at this time, but should be addressed in future studies. We have added this to the discussion (line 346).

In the hamster model, JN.1 was found to be attenuated and not transmissible, despite robust replication in hamster nasal epithelium in vivo and human nasal epithelial cells in vitro. The lack of transmission is in sharp contrast with the efficient spread of JN.1 in the human population. This points towards significant limitations of the hamster model and/or problems with the JN.1 isolate used and should be discussed in more detail and/or subjected to further investigation. For instance, did JN.1 exhibit altered aerosol stability and were mutations observed that might hint towards adaptation to the rodent host.

We agree with the reviewer that the lack of transmission of JN.1 in the hamster model is surprising. One important difference between transmission in the human population versus the hamster model is that JN.1 transmission in humans happens in the presence of pre-existing immunity against SARS-CoV-2. While JN.1 does a lot worse in the hamster transmission model compared to e.g. Delta, we can only speculate that Delta could also outcompete JN.1 during human transmission in an immunologically naïve population. In general, early during SARS-CoV-2 evolution we observed a shift towards enhanced transmission as shown for Delta in the human population (<https://doi.org/10.7554/eLife.87094.3>). Currently, the Omicron evolution is driven more by the immune landscape and escaping antibody responses than optimizing transmission efficiencies. Reduced airborne transmission of Omicron variants has been observed by several

laboratories (<https://doi.org/10.1371/journal.ppat.1010970>) which indicates this is a feature of these viruses rather than an isolate problem. Therefore, we do not think the difference is due to problems with the virus stock. We have sequenced our virus stock and did not find any consensus mutations compared to the original isolate (EPI_ISL_18563626). Moreover, we observe efficient replication of JN.1 in human nasal epithelium suggesting that there is no problem with virus replication of this isolate. Aerosol stability of several SARS-CoV-2 variants including Omicron have been tested and further indicate that aerosol stability is likely not a factor driving transmissibility (<https://doi.org/10.3201/eid2905.221752>), which is additionally supported by efficient transmission of JN.1 in the human population.

We have added a discussion of JN.1 transmission to the text (lines 310-326).

Minor

It is unclear which controls are shown in figure 2C.

The control animals in Figure 2C (now Figure 2E) and Figure 4C (now Figure 4G) are age-matched historical control animals that were not included in any of these experiments. These animals serve as reference values for comparable animals to better understand the impact of the lung/body ratio as marker for lung damage. We have clarified this in the text (lines 113, 185) and amended the figure legends.

Reviewer #2 (Remarks to the Author):

Wickenhagen et al. characterized the properties of WA-1, Delta, Omicron BA.1, XBB.1.5/1.16, EG.5.1 and JN.1 variants. The authors compared replication of these variants and induction of IFN responses in human primary epithelial cells models. They also used the hamster model to compare their pathology and their transmission, after direct contact between infected animals or through airborne transmission. They also established an antigenic mapping of the different variants. Overall, the results are interesting, providing a detailed overview of the different variants including the recent JN.1 strain, and the selective advantage. The paper is well written.

Major points:

1. The authors titrated the viral stocks on Vero E6 for the hamster experiments and on Vero E6 ACE2+/TMPRSS2+ for primary epithelial experiments. Why were two different cell lines used?

All virus stocks were titrated on VeroE6 cells and the resulting virus titers were used to calculate MOIs and infection doses for all experiments. Due to the lower virus titers observed in infected hLOs, culture supernatants were titered on VeroE6 ACE2+/TMPRSS2+ cells to increase sensitivity of the readout. We have clarified this in the manuscript (lines 526-532).

2. It has been shown that Omicron variants exhibit reduced ability to replicate in different cells such as Vero E6 or Calu-3 (Planas et al., Nature Com, 2024; Tamura et

al., Cell Host Microbe, 2024). The authors used Vero E6 to determine viral titers, which may bias the calculation of the MOI. For the hamster experiments, a figure showing the amount of inoculated RNA for each variant is needed.

VeroE6 cells were used to titrate virus stocks to avoid bias from differential TMPRSS2 usage. We acknowledge the reviewer's point that SARS-CoV-2 VOCs exhibit different replication kinetics on VeroE6 cells but TCID50 is an endpoint assay, where replication kinetics have minimal effect on the readout. Standardization of inoculum dose based on RNA copies does not differentiate between intact infectious virions, non-infectious virions, and genomic fragments, and thus does not result in a fair comparison when analyzing shedding of infectious virus. We have added the virus titer and RNA copies of each virus stock to the manuscript (lines 393-397).

3. Fig 5: The quantification of infectious virus does not seem to match the subgenomic RNA data (Fig S5). Genomic RNA should be quantified as in Figure 3.

Due to the presence of a substantial extracellular matrix layer in ihLO cultures, viral RNA can persist and be detected over a longer period of time compared to infectious virus as measured by TCID50 assay, which likely explains the presence of subgenomic RNA in the absence of infectious virus. A statistically significant increase in subgenomic RNA compared to 0hr input was only observed for variants where we also detected infectious virus (Delta, XBB.1.16). A panel showing quantification of genomic RNA in ihLOs has been added (figure S5C) and discussed in the text (line 209).

4. Fig 6: The transmission experiments do not reflect the human situation, with JN.1 being more efficiently transmitted than previous variants. Does this suggest that the in vivo hamster model is not a good model for studying the viral fitness and pathogenicity of JN.1? The role of previous immunity should be further discussed.

We agree with the reviewer that the efficient JN.1 transmission in humans is not recapitulated in an immunologically naïve hamster model. As indicated above, we suspect this difference is mainly due to pre-existing immunity towards SARS-CoV-2 in the human population. We do not think that this indicates that the hamster model is no longer useful for investigations into JN.1 pathology, but we acknowledge the limitations for transmissibility studies. We can still detect virus replication of JN.1, although to lower levels, which indicates that viral fitness can still be assessed in this model. While it is beyond the scope of this manuscript, future studies should assess viral fitness in the presence of SARS-CoV-2 immunity in the hamster model. We have additionally discussed the role of previous immunity in the manuscript to address this point (lines 310-326, line 353).

5. Fig 7: The neutralization data (ED50) could be included in an additional figure. Neutralizations were performed using Vero E6 cells using cytopathic effect as a readout for neutralizing activity of sera. As mentioned in point 1, Omicron variants, poorly replicate in these cells and are less cytopathic compared to previous variants. This presents a major concern for comparing neutralization activity against Omicron variants in Vero E6 cells. this could be further discussed as a limitation of the study.

As mentioned above in the response to point 1, we disagree with the implication that VeroE6 cells cannot be used for Omicron variants. In our hands, we still observe cytopathic effects (CPE) on these cells and can conduct assessment of virus replication on these cells. Unfortunately, we currently have no other viable alternative to assess replication competent virus titers. Therefore, we think this evaluation of neutralizing titers on VeroE6 still presents a valid comparison for Omicron variants. We have included the neutralization data as supplementary table 2.

6. Several groups compared virological and immunological properties of Omicron variants including BA.2.86/JN.1. Several references need to be added and commented in the discussion:

Tamura T, Mizuma K, Nasser H, Deguchi S, Padilla-Blanco M, Oda Y, Uriu K, Tolentino JEM, Tsujino S, Suzuki R, Kojima I, Nao N, Shimizu R, Wang L, Tsuda M, Jonathan M, Kosugi Y, Guo Z, Hinay AA Jr, Putri O, Kim Y, Tanaka YL, Asakura H, Nagashima M, Sadamasu K, Yoshimura K; Genotype to Phenotype Japan (G2P-Japan) Consortium; Saito A, Ito J, Irie T, Tanaka S, Zahradnik J, Ikeda T, Takayama K, Matsuno K, Fukuhara T, Sato K. Virological characteristics of the SARS-CoV-2 BA.2.86 variant. *Cell Host Microbe*. 2024 Feb 14;32(2):170-180.e12.

Planas D, Staropoli I, Michel V, Lemoine F, Donati F, Prot M, Porrot F, Guivel-Benhassine F, Jeyarajah B, Brisebarre A, Dehan O, Avon L, Bolland WH, Hubert M, Buchrieser J, Vanhoucke T, Rosenbaum P, Veyer D, Péré H, Lina B, Trouillet-Assant S, Hocqueloux L, Prazuck T, Simon-Lorieri E, Schwartz O. Distinct evolution of SARS-CoV-2 Omicron XBB and BA.2.86/JN.1 lineages combining increased fitness and antibody evasion. *Nat Commun*. 2024 Mar 13;15(1):2254.

Li P, Faraone JN, Hsu CC, Chamblee M, Zheng YM, Carlin C, Bednash JS, Horowitz JC, Mallampalli RK, Saif LJ, Oltz EM, Jones D, Li J, Gumina RJ, Xu K, Liu SL. Neutralization escape, infectivity, and membrane fusion of JN.1-derived SARS-CoV-2 SLip, FLiRT, and KP.2 variants. *Cell Rep*. 2024 Jul 17;43(8):114520.

Qu P, Xu K, Faraone JN, Goodarzi N, Zheng YM, Carlin C, Bednash JS, Horowitz JC, Mallampalli RK, Saif LJ, Oltz EM, Jones D, Gumina RJ, Liu SL. Immune evasion, infectivity, and fusogenicity of SARS-CoV-2 BA.2.86 and FLip variants. *Cell*. 2024 Feb 1;187(3):585-595.e6.

Zhang L, Kempf A, Nehlmeier I, Cossmann A, Richter A, Bdeir N, Graichen L, Moldenhauer AS, Dopfer-Jablonka A, Stankov MV, Simon-Loriere E, Schulz SR, Jäck HM, Čičin-Šain L, Behrens GMN, Drosten C, Hoffmann M, Pöhlmann S. SARS-CoV-2 BA.2.86 enters lung cells and evades neutralizing antibodies with high efficiency. Cell. 2024 Feb 1;187(3):596-608.e17

We have included these references as suggested (lines 44, 290, 294, 298, 311, 315, 329, 330).

7. Fig 1: Statistics are missing. The figure is difficult to read with all the bars.

We revised Figure 1 to better highlight the comparisons that are being made regarding TMPRSS2 and hamster ACE2 usage. Additionally, we have included the statistics for these graphs as suggested.

Reviewer #3 (Remarks to the Author):

The manuscript of Wickenhagen et al. adds important aspects to our understanding of SARS-CoV-2 Omicron evolution. The authors answer a relevant scientific question with a combination of in-vivo experiments in the Syrian hamster and in-vitro experiments in primary human cells of the respiratory tract. Although these experiments are interesting and valuable for the evaluation of the specific characteristics of the evaluated Omicron variants, an overall statement of SARS-CoV-2 evolution would need more controls, which are lacking in specific experiments.

We specifically have added historical hamster data for the D614G, B.1.1.7 and Delta variants in supplementary table 1 to address the concerns regarding the overall statement of SARS-CoV-2 evolution.

Specific:

I.53: Specify “non-naïve”. Assuming you refer to immunologically naïve?

Yes, here we refer to an immunologically naïve population. We have changed the wording as suggested (line 54).

II.61: What is meant with “Species susceptibility in the Syrian golden hamster”? Isn’t it just susceptibility in the Syrian golden hamster?

Changed as suggested (line 62).

II.73 and Figure 1a: Can you explain the rationale for including the hamster ACE-2, human TMPRSS2 transfected cells? Have these experiments also been performed with hamster only (either hamster ACE-2, hamster TMPRSS2 or hamster ACE-2 only)? Otherwise I wonder how conclusive those experiments are in terms of species specificity. Alternative please make clear why this selection was been made.

Furthermore, why is WA.1 used here, whereas in majority of following in-vitro experiments the authors use Delta as pre-Omicron control?

The aim of this experiment was to evaluate the dependency of different SARS-CoV-2 spike proteins on TMPRSS2 to mediate entry, as well as to compare entry efficiency with human vs hamster ACE2 since changes in spike are known to affect receptor binding. To facilitate direct comparison to human ACE2/TMPRSS2 cells, we generated hamster ACE2/human TMPRSS2 cells. Due to the reduced TMPRSS2 usage of the Omicron variants, we did not generate a hamster TMPRSS2-expressing cell line. We have further highlighted this in the text and clarified the rationale behind this experiment (lines 73-93). Additionally, we have now added Delta as a control in these experiments as suggested and changed the presentation of the data to increase readability (Figure 1).

II.82: Although there is an increase for Omicron EG.5, later variants dip again in binding. It seems there is no clear dynamic with the emergence of variants over time. Any explanations or thoughts how that mechanism than influences evolution?

The main comparison we are making is the TMPRSS2 and hamster ACE2 usage within each variant pseudotype. We see a general trend towards reduced TMPRSS2 dependency of Omicron variants in comparison to Delta and WA-1 (Fig. 1A). As indicated in response to reviewer 1, our experiments are not ideally designed to compare the different variants and we hesitate to draw conclusions based on some of the smaller inter-variant differences we observed. We have modified the text (lines 73-93) to better highlight the comparisons we are making and have also revised figure 1 to increase readability.

II.85: I agree that incorporation of spike is overall successful, yet variable. Can that have an influence on this experiment?

We agree with the reviewer that we observed variable spike incorporation in our assay. But we think that differential incorporation of spike is not a concern as only intra-pseudotype comparisons are relevant. We are comparing the same pseudotype stock on cells expressing different ACE2 and/or TMPRSS2 proteins. Therefore, differential incorporation of spike should not influence the overall comparisons drawn within each pseudovirus. We have clarified this in line 86-89.

II.97: Authors state there is no weight loss, but what about animals inoculated with EG.5 at 2 dpi? An equivalent loss of body weight is interpreted as body weight loss in the following experiment (intratracheal inoculation) of the authors.

The animals inoculated I.N. with EG.5.1 at 2 DPI have a mean weight of 99.2%, compared to 97.7% for I.T.-inoculated hamsters at 5 DPI. The weight change for I.N. hamsters represents a <1g weight differential on a 90-100g hamster and is likely attributed to daily fluctuation or stuffing additional food into their pouches.

We have equalized the axes of these graphs (Figure S2A, S4A) to facilitate comparison between I.N. and I.T. inoculations.

Overall for the experiment of intranasal inoculation of hamsters I am missing a pre-Omicron control group to assess if there is indeed the evolutionary trend towards less LRT disease and replication and especially more efficient replication in the URT. Maybe historical controls of previous experiments can be incorporated or the interpretation of data should be tuned down (focus on intra-Omicron description, not general SARS-CoV-2 evolution).

We have addressed this comment by adding historical hamster data for clade B ancestor D614G, B.1.1.7(alpha), and Delta variants in supplementary table 1. This historical data was produced in our lab using age- and sex-matched hamsters obtained from the same vendor and thus is directly comparable to the experiments in this manuscript. The publications pertaining to this historical data are referenced in the table. These additional data combined with our experiments indicate the trend towards less LRT disease. We have further discussed this in the text (lines 105-107, 174-176).

LI.143: This was also briefly mentioned in your discussion, but this sentence is an overinterpretation to me. Infection of ALI cultures with SARS-CoV-2, independent of which variant chosen, barely induces any cpe. This stands in sharp contrast to observations in the field and in-vivo. Therefore the absence of cpe should not be used as argument for loss of LRT pathology. Especially since the same absence is observed for Delta.

We have modified this statement in the text (line 149) to clarify that enhanced replication capacity in combination with the lack of increased cell death is suggestive of adaptation to the upper respiratory tract. Our discussion of lower respiratory tract pathogenicity is based on virus replication data as opposed to cytotoxicity data, as the reviewer correctly notes that cytotoxicity is not an accurate metric of pathogenicity in the lung, given that Delta is not cytolytic in this model. We have modified the text to clarify this (line 217).

LI.146: It would be important to hear the rationale of the authors to conduct this inoculation route. What is the relevance and how can results be translated to observations in the field? What is the added value of this elaborate in-vivo experiment?

Early Omicron variants displayed an attenuated disease phenotype in several animal models. Initially, it was hypothesized that reduced replication due to altered entry of the virus resulted in the absence of pathology in the lower respiratory tract (LRT). We previously showed that this attenuated disease phenotype can be overcome by directly depositing the virus into the LRT via intratracheal inoculation resulting in efficient replication and similar pathology in the LRT compared to earlier variants (<https://doi.org/10.1038/s44298-023-00012-2>).

This indicates that virus replication in the lower respiratory tract itself is not the barrier to the reduced pathology observed after intranasal inoculation. Continued assessment of antiviral treatments and vaccines all require the presence of pathology which can be achieved using intratracheal inoculation. Based on this rationale we wanted to investigate if contemporary Omicron variants were also able to replicate and cause pathology in the lower respiratory tract after intratracheal inoculation in order to assess whether the hamster model would facilitate evaluation of therapeutic and vaccine efficacy against these variants. Furthermore, we observe important phenotypic differences where JN.1 deposited intratracheally struggles to migrate into the upper respiratory tract despite replicating in the lower respiratory tract, while XBB and EG.5.1 variants migrated into the upper respiratory tract after intratracheal inoculation. We have further discussed this in the manuscript (lines 155-162).

Il.180: For the reading flow, it would be better if the last paragraph would follow upon Il. 168.

Changed as suggested (lines 110-121, Figure 2, 184-199, Figure 4)

Il.195: Interesting to see that XBB.1.16 infected hLOs produce highest amount of infectious virus. In hamsters you see most of pathology here, yet not the highest amount of infectious virus (or RNA). Would be interesting to elaborate on those observations in more detail in the discussion. What can cause these differences?

Taken together, greater ISG induction in hLOs along with higher histopathology scores in hamsters suggests increased immune activation in response to XBB.1.16-infection. *In vivo*, this immune activation may drive pathology, but also limit virus replication. The differences in virus replication trends between ihLOs and hamsters may be a result of species differences, or differences in cellular composition: specialized immune cells that effectively control virus replication *in vivo* are missing in ihLOs. We have expanded on this point in the revised manuscript (lines 337-339).

Il.207: Can you visualize the comparison of LRT vs URT? Since this is a very relevant observation/question, it would be helpful to have a comprehensive figure indicating comparative findings.

We have added a panel (Figure 5C) directly comparing viral sgRNA in ihLOs and nasal ALI cultures.

Il.210: This is a very strong statement, which is not underlined with the needed data since comparisons rely on 1 pre-Omicron variant only.

We have changed the text to highlight that this comparison is done with Delta and weakened the implication that this is true for all variants pre-Omicron (line 225).

II.218: What is the rationale to include this specific Omicron variant? Especially since BA.1 was not focus of previous experiments?

For the transmission experiment BA.1 was chosen as comparison to better highlight intra-Omicron variation and because previous transmission data on BA.1 was available which indicated that BA.1 would achieve contact transmission and limited airborne transmission (<https://doi.org/10.1038/s44298-023-00012-2>). This allowed us to assess if an increase in airborne transmission in later Omicron variants was present or not. We have clarified this in the text (lines 233-235).

II.241: Why now Alpha and WA.1? Generally the inclusion of different variants seems to be inconsistent throughout the different experiments.

The inclusion of WA-1 and Alpha in this assay was chosen to have antigenically distinct variants for the generation of the antigenic map and to capture the breadth of SARS-CoV-2 evolution. We have clarified this in the text (line 257).

II.297: Is this than an inherent feature of the virus or just linked to virus titers (excl. Delta)?

Due to the lack of virus replication in ihLOs infected with Omicron variants (with the exception of XBB.1.16), we cannot make definitive conclusions regarding ISG induction in hLOs. We hypothesize that the lack of ISG induction is a result of the absence of virus replication due to the following: (1) we observe strong ISG induction in Omicron-infected nasal ALI cultures, and (2) in a prior study we have observed ISG induction in Omicron BA.1-infected lung organoids in one donor where virus replication was also observed (ref. 13). This is mentioned in lines 270, 273, and 340-342 of the revised manuscript.

II.307: Not only the transmission experiments do not consider pre-existing immunity. Also pathogenesis in-vivo experiments and infections of primary cells does not.

We agree with the reviewer that this is a caveat. The goal of this study was to compare the virus-intrinsic differences in replication, pathogenicity, and transmission. Going forward, we think the assessment of transmission and pathogenesis in the hamster model for SARS-CoV-2 should be considered in the presence of previous immunity. We have added a discussion of the impact of pre-existing immunity on transmission (lines 310-326), as well as highlighted the limitations of the human primary cell culture and immunologically naïve hamster model systems (lines 350-357).

Figure 1: Although the authors show that Omicron is less dependent on TMPRSS2, can the human TMPRSS2 outweigh the less efficient entry via hamster ACE-2? Have statistics been performed here?

We have added statistics to Figure 1 and included an additional comparison looking at the efficiency of the hamster ACE2 entry over human ACE2 (Fig. 1B). While we can not exclude an effect of TMPRSS2 masking the hamster ACE2, we do show a reduced dependency of the Omicron variants on TMPRSS2 and therefore believe this is unlikely.

Figure 2: Results match with observations of experiments visualized in figure 1 for results in the lungs, not so much for the nose. Any ideas why?

Experiments in figure 1 evaluate the impact of TMPRSS2 and human vs hamster ACE2 usage on pseudovirus entry efficiency of different Omicron spikes, and we have highlighted the limitations of comparing entry efficiency between variants in the revised manuscript. Virus replication *in vivo* can be impacted by multiple additional factors such as the immune response, which can differ between the upper and lower respiratory tract. Additionally, direct deposition of virus into the nose with I.N. inoculation likely impacts virus replication in that site.

Figure 4: How do the authors explain the significant differences observed in IHC, but not in viral titration (4D vs 4I) at 3 dpi?

These differences can be explained by the difference in techniques. Virus titers are determined on one piece of tissue and are quantitative, displayed on a log scale. In comparison, IHC takes the whole lung into account but is a semi-quantitative method. Different techniques with different scales and different data are very hard to compare.

Reviewer #1 (Remarks to the Author):

Minor: "In this study we compared virus-intrinsic transmission efficiency in immunologically naïve hamsters and observed reduced transmissibility of JN.1 compared to BA.1. Future studies should investigate how prior immunity impacts both transmissibility and pathogenesis of JN.1". This statement is not fully clear to this reviewer. JN.1 transmissibility was not only reduced, it was absent. Is a different situation expected in immunologically non-naive animals?

The reviewer is correct that the true transmissibility of JN.1 cannot be evaluated, however some of our previous studies have shown that prior immunity (either from vaccine or previous SARS-CoV-2 exposure) would select for the most antigenic distant variant (<https://www.nature.com/articles/s41467-023-42346-8>). And we can then deduce the impact of prior immunity (e.g. derived by antigenic distant strains), by doing co-infections with the older outgoing strain and the newer strain to investigate the selective advantage in the context of prior immunity.

We have adapted the wording in line 324 of the manuscript to highlight the absent transmission of JN.1.